# Recent range expansion and lineage idiosyncratic population structure of *Liodessus* diving beetles in the high Andes (Coleoptera: Dytiscidae, Bidessini)

**Michael Balke**[1,2], **Tobias Mainda**[3], **Katja Neven**[1], **Lars Hendrich**[1], **Michael Steven Basantes**[4], **Carlos Prieto**[5,6], **Adrián Villastrigo**[1]*

**1** SNSB-Zoologische Staatssammlung München, Munich, Germany, **2** GeoBioCenter, Ludwig Maximilians University, Munich, Germany, **3** Zoological Institute and Museum, University of Greifswald, Greifswald, Germany, **4** Entomology Division, Instituto Nacional de Biodiversidad INABIO, Quito, Ecuador, **5** Departamento de Biología, Universidad del Atlántico, Barranquilla, Colombia, **6** Corporación Universitaria Autónoma del Cauca, Popayán, Colombia

* adrianvillastrigo@gmail.com

**Editor:** Łukasz Kajtoch, Institute of Systematics and Evolution of Animals Polish Academy of Sciences, POLAND

**Data Availability Statement:** The alignments presented here have been deposited in the Open

## Abstract

Here, we review the taxonomy and population genetic structure of diving beetles in the genus *Liodessus* Guignot, 1939 from the high Andes of southern Colombia and Ecuador. *Liodessus quillacinga ecuadoriensis* ssp. nov. is described from the type locality Otavalo, Laguna San Pablo. *Liodessus quimbaya azufralis* Megna, Hendrich & Balke, 2019 stat. nov. is now regarded as a subspecies of *Liodessus quimbaya* Megna, Hendrich & Balke, 2019 based on new morphological and genetical data. *Liodessus quimbaya paletara* ssp. nov. is described from the Paletará Valley (Colombia: Cauca).

## Introduction

The Northern Andes have a diverse topography, characterized by strong climatic disparities across a relatively small region [1, 2]. The vast, isolated high-altitude area of the Northern Andes, with its large intramontane depressions, is covered by the unique tundra-alpine Páramo ecosystem between latitudes 11˚N and 8˚S. Páramos extend over an area of 35,000 km² at altitudes between 2,800 and ca. 4,700 m [3]. Already marvelled by von Humboldt [4], these regions are now acknowledged as botanical biodiversity hotspots, hosting around 5,000 almost exclusive endemic plant species [5]. In the Quaternary, the Colombian Páramos have been subjected to a dynamic series of connection and disconnection events driven by alternating glacial and interglacial periods, which is referred to as a "flickering connectivity system" [2]. This intricate process has not only impacted the interconnectivity of major mountain ranges, but has also influenced the connectivity of Páramo areas. Consequently, it led to complex speciation processes shaped by the dynamic interactions of populations and their dispersal capabilities. Furthermore, the reduction in wing size, a common insect adaptation to high elevation [6], may curtail dispersal capabilities. Recent studies indicate a complex

1 / 30

Science Framework repository (https://doi.org/10.17605/OSF.IO/3P8QR). The locality and sequence data are also available in the Barcode of Life Data System (BOLD, https://www.boldsystems.org) in the public project COLLI.

**Funding:** Michael Balke acknowledges support from the Deutsche Forschungsgemeinschaft (BA2152/27-1), project number: 496550039. In the initial phases of this project, Adrián Villastrigo was funded by the Alexander von Humboldt Foundation through a Humboldt Research Fellowship and by the Carl Friedrich von Siemens Foundation at SNSB-ZSM, and this stipend is greatly acknowledged here. This work was made possible by a grant from the Alexander von Humboldt-Stiftung under the Research Group Linkage Program (Evolution of the high Andean insect fauna project) to Michael Balke and Carlos Prieto. Michael Balke acknowledges support from the EU SYNTHESYS program, projects FR-TAF 6972 and GB-TAF-6776, which supported this research during visits to Natural History Museum in London and Muséum national d'Histoire naturelle in Paris in 2017 to study historical type material.

**Competing interests:** The authors have declared that no competing interests exist.

interplay of biotic and extrinsic factors, along with diversity-dependent processes, which could determine the shaping of diversity patterns and the distribution of mountain species, e.g. [7–9]. Notably, recent attention has been devoted to high Andean diving beetle communities, particularly those belonging to the genus *Liodessus*, Guignot, 1939, e.g. [10].

*Liodessus*, a genus of the tribe Bidessini (Dytiscidae, Hydroporinae), contains 50 beetle species characterized by their diminutive size–predominantly smaller than 3 mm, and their distribution in the Neotropical and Nearctic regions [11]. It is noteworthy that Afrotropical species currently assigned to this genus [12] represent in fact a different, not closely related lineage (Michael Balke, personal communication). Within the Andes, *Liodessus* are found from lowlands to nearly 5,000 m, primarily inhabiting lotic habitats (rarely slow-flowing channels) where they often constitute the most abundant aquatic beetles [13, 14]. Despite their prevalence, *Liodessus* from the high-altitude Páramo and Puna ecoregions remained poorly studied. Over the past few years, numerous reports on *Liodessus* from these regions have been published, including descriptions of new taxa and documentation of new records, e.g. [10, 13–16]. To support and enhance morphological results, and better understand distribution patterns and phylogenetic affinities of *Liodessus* taxa, the public COLLI project ("Colombian and Andean diving beetles in the genus *Liodessus*", see [17] for details) was initiated, based on the Barcode of Life Data System (BOLD) of the Canadian Centre for DNA Barcoding, specifically targeting the 5' end of the mitochondrial cytochrome *c* oxidase subunit I (5-COI) gene fragment [18]. COLLI currently contains 466 sequences of 5-COI for more than 30 taxa, and these numbers are constantly growing due to the very active nature of this database, with ongoing additions of sequence data and taxonomic curation.

To date, only *Liodessus riveti* Peschet, 1923 [19] was reported from Ecuador (see below). On the other hand, more attention has been focused on the fauna of the Colombian Andes, with six species plus seven subspecies currently recognized [10]. The same is true for the Andean fauna of Peru, south of Ecuador, with seven described species [11].

Balke et al. [10] introduced an updated interpretation of the *Liodessus* fauna of the eastern branch of the Colombian Andes, suggesting the *L. bogotensis*-complex, for populations widespread along this section of the Andes. This complex now contains various subspecies introduced to account for genetic structure and in some cases subtle morphological differentiation related to different Páramo or Altiplano areas (geographic structure). In alignment with this methodology, we combined morphological analyses with 5-COI sequencing to present the first comprehensive assessment of the high Andean aquatic beetle fauna in northern Ecuador and provide a more comprehensive assessment of the (southern) Colombian fauna. The specific objectives are (1) taxonomic identification, (2) re-assessment of described taxa, (3) providing a refined geographic interpretation and (4) the revealing of the population genetic structure within the studied taxa.

## Material and methods

### Sampling and map generation

New samples were acquired in the framework of a large-scale Andean water beetle research project (headed by Michael Balke in cooperation with researchers from across south America). In Ecuador, new collections were obtained by Michael Basantes (INABIO, Quito) and Michael Balke; in Colombia by Carlos Prieto (Universidad del Atlántico, Barranquilla). A map was generated using QGIS v3.16.4 (www.qgis.org) using the Natural Earth raster map as base layer. Locality information and species lineage (sensu molecular information, see below) was represented by using a categorized visualization proportional to the sample size in the map.

The international cooperations underlying this project were formalized with MoUs, based on which we acquired all necessary collecting, export and DNA sequencing permits. These are archived at SNSB-ZSM and available upon request from Michael Balke. The material collected in Ecuador is supported by the permit "Diversity distribution and evolution of the beetles of Ecuador—multidisciplinary approach with emphasis on systematics genetics and biogeography" (MAATE-DBI-CM—2022-0255) issued by the Ministry of Environment, Water and Ecological Transition of Ecuador. The Colombian authority ANLA (Autoridad Nacional de Licencias Ambientales) supported our expedition under the frame permit 00594.

## Acronyms

INABIO Instituto Nacional de Biodiversidad, Quito, Ecuador,

LIAUN Laboratorio de Insectos Acuáticos, Departamento de Biología, Universidad Nacional de Colombia,

UNAL Colombian National Collecion of Entomology, ICN, Universidad Nacional de Colombia,

ZSM SNSB-Zoologische Staatssammlung, Munich, Germany, vouchers temporarily stored for further morphological work.

Codes in the format ECU_MB_2022_01 in the studied material sections are ZSM locality codes, and refer to the country of origin (ECU = Ecuador; COL = Colombia), collector who organized the fieldwork (cPr = Carlos Prieto, MB = Michael Balke, MSB = Michael Steven Basantes, YSM = Yoandri Suarez Megna), year of collection (e.g., 2022) and a locality number for the respective collecting event (e.g., 01).

## Morphological descriptions and photography

A minimalistic description format of morphological characters follows previous publications on *Liodessus*, e.g. [14]. Images were taken using a Canon EOS R camera. A Mitutoyo 10x ELWD Plan Apo objective was used for the habitus, and a corresponding 20x objective for the genital structures. These objectives were attached to a Carl Zeiss Jena Sonnar 3.5/135 MC as focus lens. Three SN-1 LED segments from Stonemaster were used for illumination (www. stonemaster-onlineshop.de). Image stacks were created using a Stackmaster macro rail (Stonemaster). The stacks were fused using Helicon Focus v. 7.61 on a MacPro 2019 (Apple Inc.) with a Radeon Pro 6800X MPX GPU.

## DNA sequencing

The laboratory protocol for DNA sequencing and subsequent data analysis follows the Canadian Centre for DNA Barcoding (CCDB) standard procedures for barcoding (ww.ccdb.ca). Tissue samples were processed at CCDB, with the resulting barcode sequences (COI-5) being uploaded to the BOLD-system. A visual inspection of the chromatograms as provided by BOLD-system was done to assess for sequence correctness. To assess the relationship between newly acquired sequences and existing species, a straightforward approach was followed: a neighbour-joining tree, based on *p*-distances estimated by the Jukes-Cantor model, was computed using Geneious v11.0.4 [20]. This methodology has been used in recent studies on *Liodessus* (e.g. [10]) to support the description process based on morphological data (in combination with molecular and geographical evidence).

## Trees, haplotype networks and the calculation of genetic distance

Nucleotide sequences of 206 specimens, belonging to the taxa referred above, were retrieved from BOLD-system and subjected to alignment using MAFFT v7.450 [21] with the AUTO

algorithm, as implemented in Geneious v11.04 software. For quality control, the nucleotide alignments were translated into amino acid sequences, allowing for visual inspection to identify potential sequencing errors, specifically internal stop codons. This step served as a crucial measure to ensure the accuracy and reliability of the data. To better comprehend the information content embedded in the nucleotide data, the Geneious tree builder was employed by using the neighbour-joining method with the Jukes Cantor genetic distances. Nucleotide DNA alignments were used to estimate haplotypes using PHASE [22] through 1,000 iterations. A burn-in fraction of 10% and a thinning parameter of 5 were applied during this process to ensure robust results. Subsequently, an alternative representation of the relationship of the genetic information was performed by using haplotype networks constructed using TCS software [23], considering a connection threshold of 95% to generate a fully connected network. For enhanced visualization of the haplotype network, tcsBU [24] was used considering collecting sites used as groups. Pairwise genetic differentiation among populations was explored in DNAsp v6.12.03 [25]. The fixation index (Fst) was calculated as a metric for assessing connectivity between different populations. Visualization of the Fst values was represented as a heatmap using Python 3 [26] complemented by the Seaborn library [27].

## Diagnostic characters

The aligned nucleotide sequences were analysed for each species in Geneious v11.04 using the "Highlighting > All disagreement to consensus" function. The diagnostic characters were visually inspected to identify unique consensus bases whose position in the multiple sequence alignment for each of the sublineages.

## Species delimitation methods

To further explore genetic diversity and validity species boundaries, we employed two commonly used species delimitation methods: Assemble Species by Automatic Partitioning (ASAP [28]) and the Poisson Tree Process (PTP [29]). These methods are particularly effective in addressing the challenges of cryptic species and complex genetic structures, although they can be sensitive and unreliable when gene flow is present [30]. For the ASAP method, we used the nucleotide alignments with the Kimura-2P model, splitting groups with a probability below 0.01. For PTP, we use a Maximum Likelihood tree generated by IQ-Tree v.2.3.2 [31] with default settings and 10,000 bootstrap replicates. The PTP analysis ran for 100,000 generations, with a burn-in fraction of 10% and a thinning parameter of 100.

## Nomenclatural acts

The electronic edition of this article conforms to the requirements of the amended International Code of Zoological Nomenclature, and hence the new names contained herein are available under that Code from the electronic edition of this article. This published work and the nomenclatural acts it contains have been registered in ZooBank, the online registration system for the ICZN. The ZooBank LSIDs (Life Science Identifiers) can be resolved and the associated information viewed through any standard web browser by appending the LSID to the prefix ""http://zoobank.org/"". The LSID for this publication is: urn:lsid:zoobank.org:pub:296109F0-6DEE-4480-9B4B-DAE67A047596. The electronic edition of this work was published in a journal with an ISSN, and has been archived and is available from the following digital repositories: PubMed Central and LOCKSS.

# Results

## Taxonomy

***Liodessus quillacinga quillacinga*** Megna, Hendrich & Balke, 2019. Figs 1, 2, 3A, 4A, 5B, 6 and 7

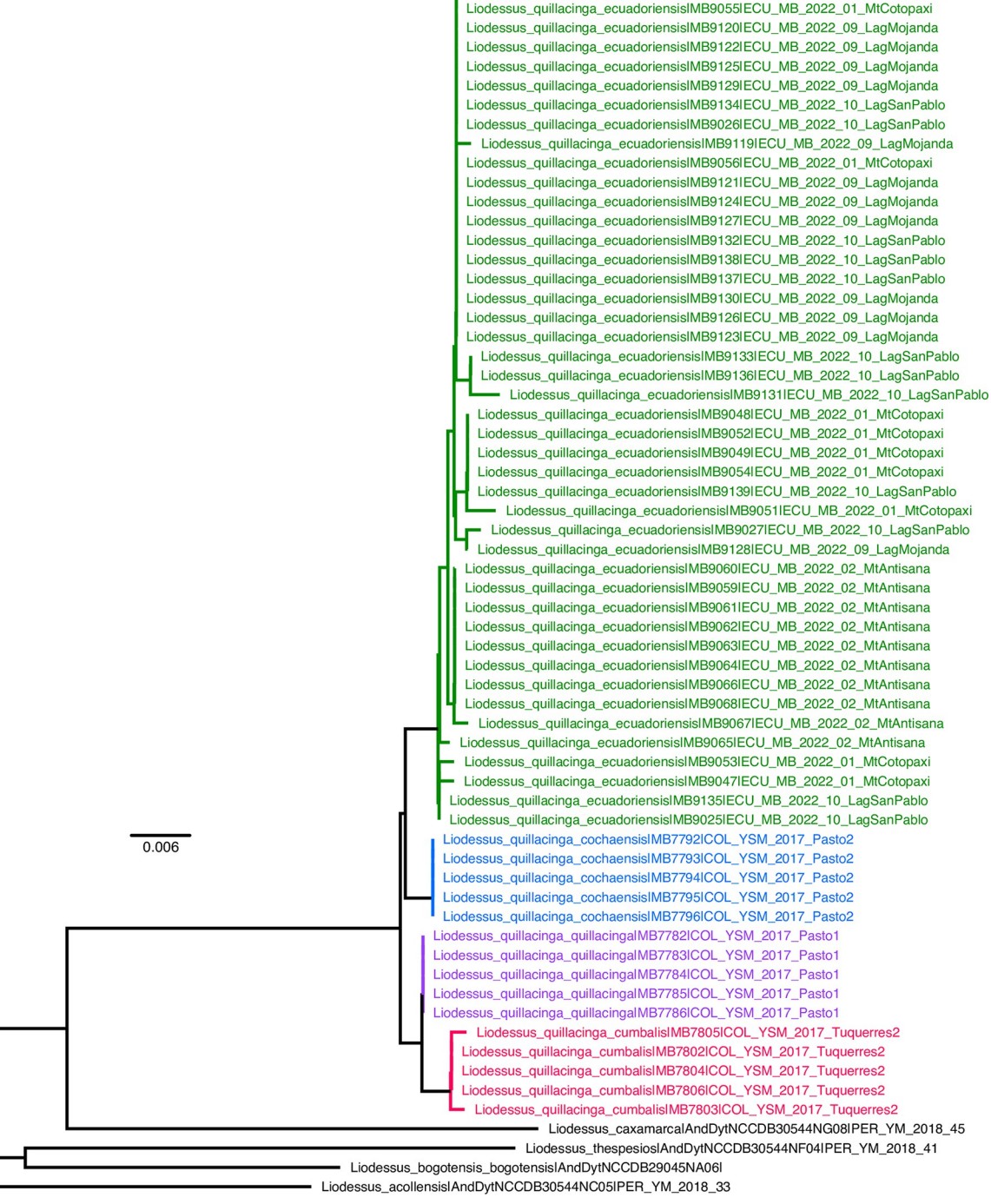

**Fig 1. Neighbour-joining tree for the sublineages of *Liodessus quillacinga*. Green** *L. quillacinga ecuadoriensis* ssp. nov., **Blue** *L. quillacinga cochaensis*, **Pink** *L. quillacinga quillacinga*, **Red** *L. quillacinga cumbalis*, **Black** outgroup.

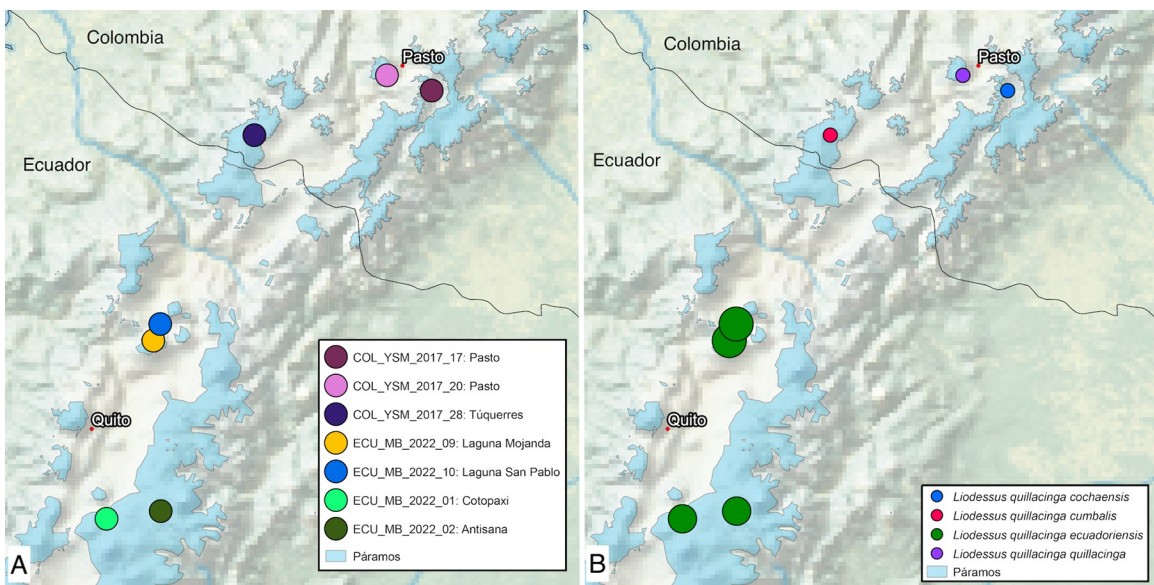

**Fig 2.** Maps of **A** sampling localities and **B** lineages of *Liodessus quillacinga*. Blue coloured areas represent Páramo habitats.

**Type locality.** Colombia: Nariño, Pasto, surroundings of the Galeras volcano, 1.166˚ -77.333˚. **Holotype.** Colombia: Nariño, Pasto, surroundings of the Galeras volcano, 3,200 m, 05.v.2017, 1.166˚ -77.333˚, Megna & Ruíz (CO-20) (LIAUN). **Paratypes.** See Megna et al. [16].

**Description of holotype.** Total length 2.0 mm; length without head 1.7 mm; maximum width 0.9 mm. Shape elongate, oval with lateral outline discontinuous between pronotum and elytron. Pronotum broadest in its midlength, sites convex. Elytra widest at about midlength.

**Colour.** Dorsally and ventrally testaceous to dark reddish-brown. Antenna, mouthparts, pro-, meso- and metathoracic legs testaceous (Fig 3A).

**Surface sculpture.** Head with distinct microreticulation and few punctures. Pronotum shiny, anteriorly with faint microreticulation; with dense unevenly distributed coarse setiferous punctation. Elytron shiny; surface with dense and coarse setiferous punctation.

**Structures.** Head with distinct occipital line. Pronotum with lateral bead; with distinct and very deep basal striae. Elytron with short and deep basal striae.

**Genitalia.** Median lobe of aedeagus in lateral view simply curved, broadened medially, in ventral view lateral margins parallel basally and strongly narrowed.

**Metathoracic wings.** Vestigial, reduced to short membranous stubs without sign of venation.

**Variation.** (*N* = 11) Total length 2.0–2.2 mm; length without head 1.7–1.9 mm; maximum width 0.9–1.0 mm. The extent of the basal elytral striae varies from slightly to well expressed. The dorsal coloration can be slightly darker, laterally on pronotum and elytron in some specimens. The structure of median lobe of the aedeagus in lateral and ventral view is constant in general shape as illustrated in Figs 4A and 5B.

**Female.** Dorsal side dull, with strong microreticulation.

**Identification notes.** In the COLLI sequence database, *L. quillacinga quillacinga* has two diagnostic characters different from the other sublineages of *L. quillacinga* (Table 1). The haplotype network structure is depicted in Fig 6.

**Distribution.** To date only known from its type locality in the Pasto Massif in southern Colombia at an altitude of 3,200 m (Fig 2A, 2B).

**Habitat.** Unshaded permanent lagoon with aquatic vegetation (Fig 10B Megna et al. [16]).

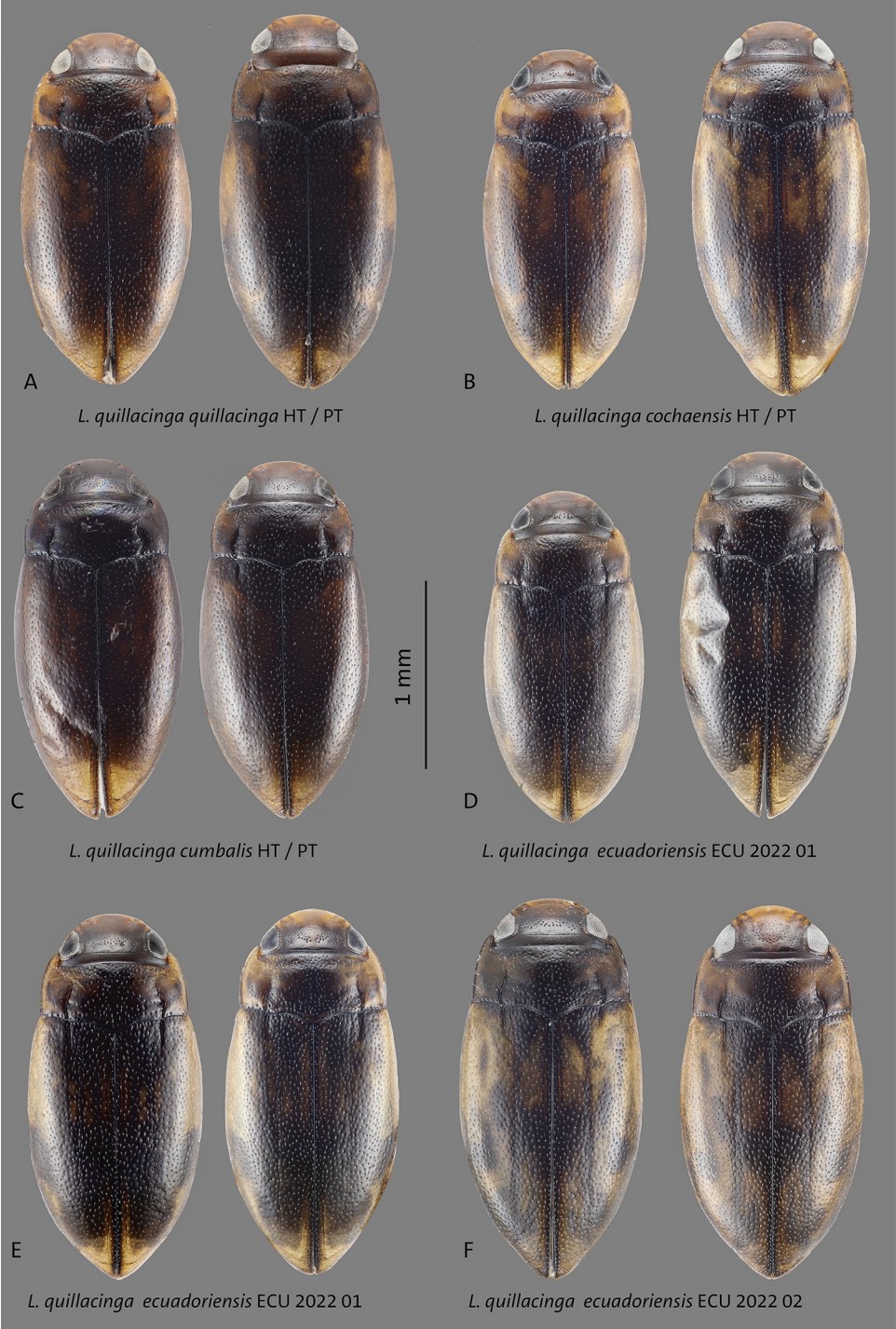

**Fig 3.** Habitus of **A** *L. quillacinga quillacinga* holotype (HT) and paratype (PT), **B** *L. quillacinga cochaensis* holotype (HT) and paratype (PT), **C** *L. cumbalis* holotype (HT) and paratype (PT), **D–F** *L. quillacinga ecuadoriensis* ssp. nov. paratypes from different localities. Scale bar: 1 mm.

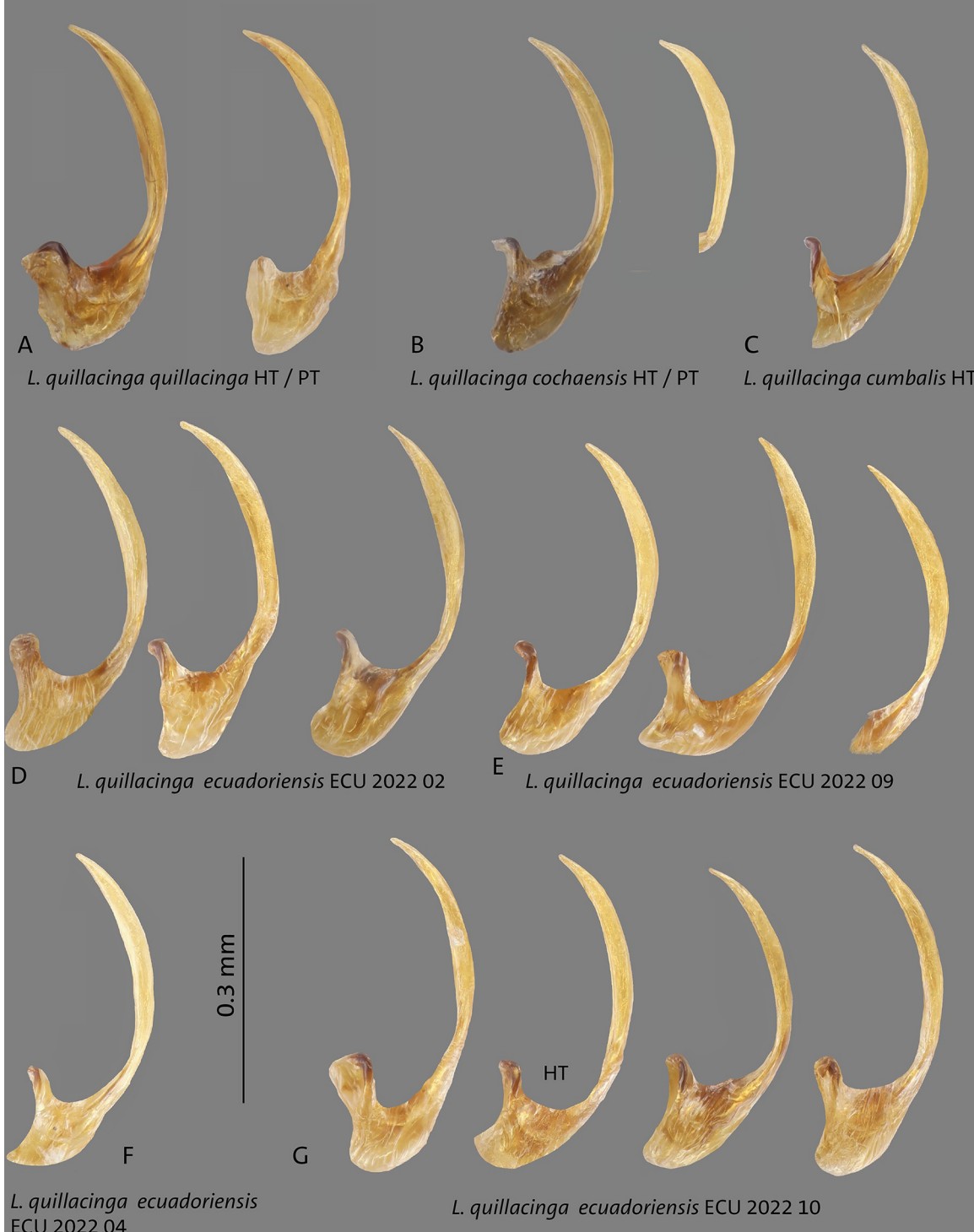

**Fig 4.** Median lobes in lateral view of *Liodessus* species **A** *L. quillacinga quillacinga* holotype (HT) and paratype (PT), **B** *L. quillacinga cochaensis* holotype (HT) and paratype (PT), **C** *L. cumbalis* holotype (HT) and **D–G** *L. quillacinga ecuadoriensis* ssp. nov. paratypes from different localities. Scale bar: 0.3 mm.

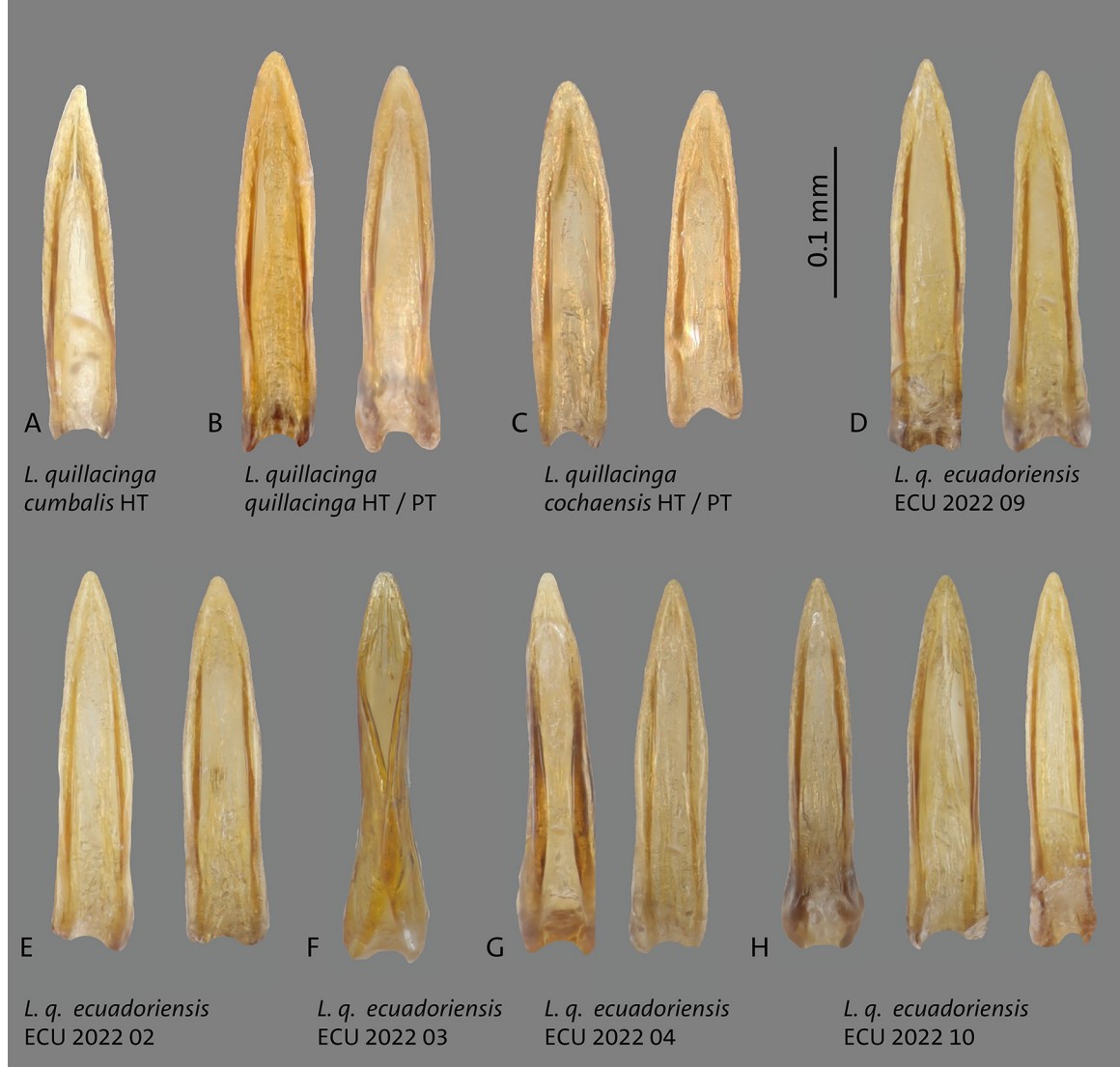

**Fig 5.** Median lobes in ventral view of *Liodessus* species **A** *L. quillacinga cumbalis* holotype (HT), **B** *L. quillacinga quillacinga* holotype (HT) and paratype (PT), **B** *L. quillacinga cochaensis* holotype (HT) and paratype (PT), **D–H** *L. quillacinga ecuadoriensis* ssp. nov. paratypes from different localities. Scale bar: 0.1 mm.

*Liodessus quillacinga cochaensis* **Megna, Hendrich & Balke, 2019.** Figs 1, 2, 3B, 4B, 5C, 6 and 7

**Type locality.** Colombia: Nariño, Pasto, Vereda El Motilón, Cocha lagoon, 1.116° -77.166°. **Holotype.** Colombia: Nariño, Pasto, Vereda El Motilón, Cocha lagoon, 2,790 m, 02. v.2017, 1.166° -77.166°, Megna & Ruíz (CO-17) (LIAUN). **Paratypes.** See Megna et al. [16].

**Size of holotype.** Total length 1.9 mm; length without head 1.6 mm; maximum width 0.9 mm. **Variation.** ($N$ = 9) Total length 1.9–2.2 mm; length without head 1.6–1.9 mm; maximum width 0.9–1.0 mm.

**Metathoracic wings.** Fully developed.

**Diagnosis.** This subspecies is very similar to other populations, it may be separated from *L. quillacinga quillacinga* and from *L. quillacinga cumbalis* by having irregular maculations on the elytra (Fig 3B). From *L. quillacinga ecuadoriensis* ssp. nov., it may be separated by the

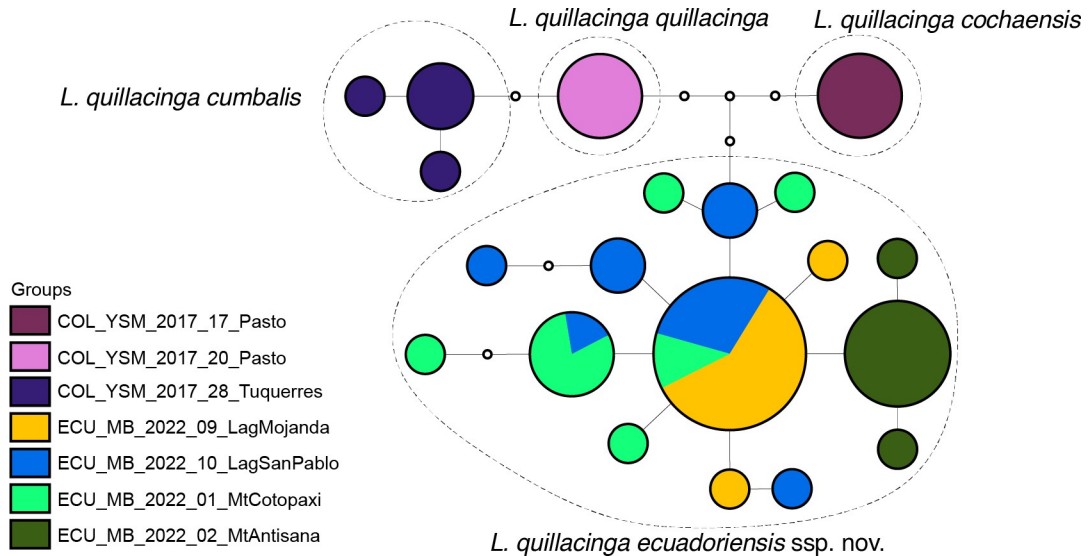

**Fig 6. Haplotype network for *Liodessus quillacinga*, including information of main sublineages.**

median lobe of aedeagus (Figs 4B and 5C). This subspecies is characterized geographically, known only from its type locality in southern Colombia. In the COLLI sequence database, *L. quillacinga cochaensis* has two diagnostic characters different from the other sublineages (Table 1). The haplotype network structure is depicted in Fig 6.

**Distribution.** To date only known from its type locality in the Pasto Massif of southern Colombia at an altitude of 2,790 m (Fig 2A and 2B).

**Habitat.** Permanent, exposed lagoon with aquatic plants.

***Liodessus quillacinga cumbalis* Megna, Hendrich & Balke, 2019.** Figs 1, 2, 3C, 4C, 5A, 6 and 7

**Type locality.** Colombia: Nariño, Túquerres, surroundings of the Cumbal volcano, Vereda Qulismar, 1.091˚ -77.718˚. **Holotype.** Colombia: Nariño, Túquerres, surroundings of the Cumbal volcano, Vereda Qulismar, 3,840 m, 10.v.2017, 1.091˚ -77.718˚, Megna & Ruíz (CO-28) (LIAUN). **Paratypes.** See Megna et al. [16].

**Size of holotype.** Total length 2.0 mm; length without head 1.7 mm; maximum width 0.9 mm. **Variation.** ($N$ = 4) Total length 2.0–2.2 mm; length without head 1.7–1.9 mm; maximum width 0.9–1.0 mm.

**Metathoracic wings.** Fully developed.

**Diagnosis.** This subspecies is very similar to other populations, it may be separated from *L. quillacinga quillacinga* and from *L. quillacinga cumbalis* by having generally darker elytra with less maculations (Fig 3C). From *L. quillacinga ecuadoriensis* ssp. nov., it may be separated by the median lobe of aedeagus (Figs 4C and 5A). This subspecies is characterized geographically, known only from its type locality in southern Colombia. In the COLLI sequence database, *L. quillacinga cumbalis* has four diagnostic characters different from the other sublineages (Table 1). The haplotype network structure is depicted in Fig 6.

**Distribution.** To date only known from the surroundings of the Cumbal volcano in the Pasto Massif of southern Colombia at an altitude of 3,840 m (Fig 2A, 2B).

**Habitat.** Páramo bog with muddy soil and submerged vegetation.

***Liodessus quillacinga ecuadoriensis* ssp. nov.** Figs 1, 2, 3D–3F, 4D–4G, 5D–5H, 6, 7, 8A–8D, 9, 10A–10C

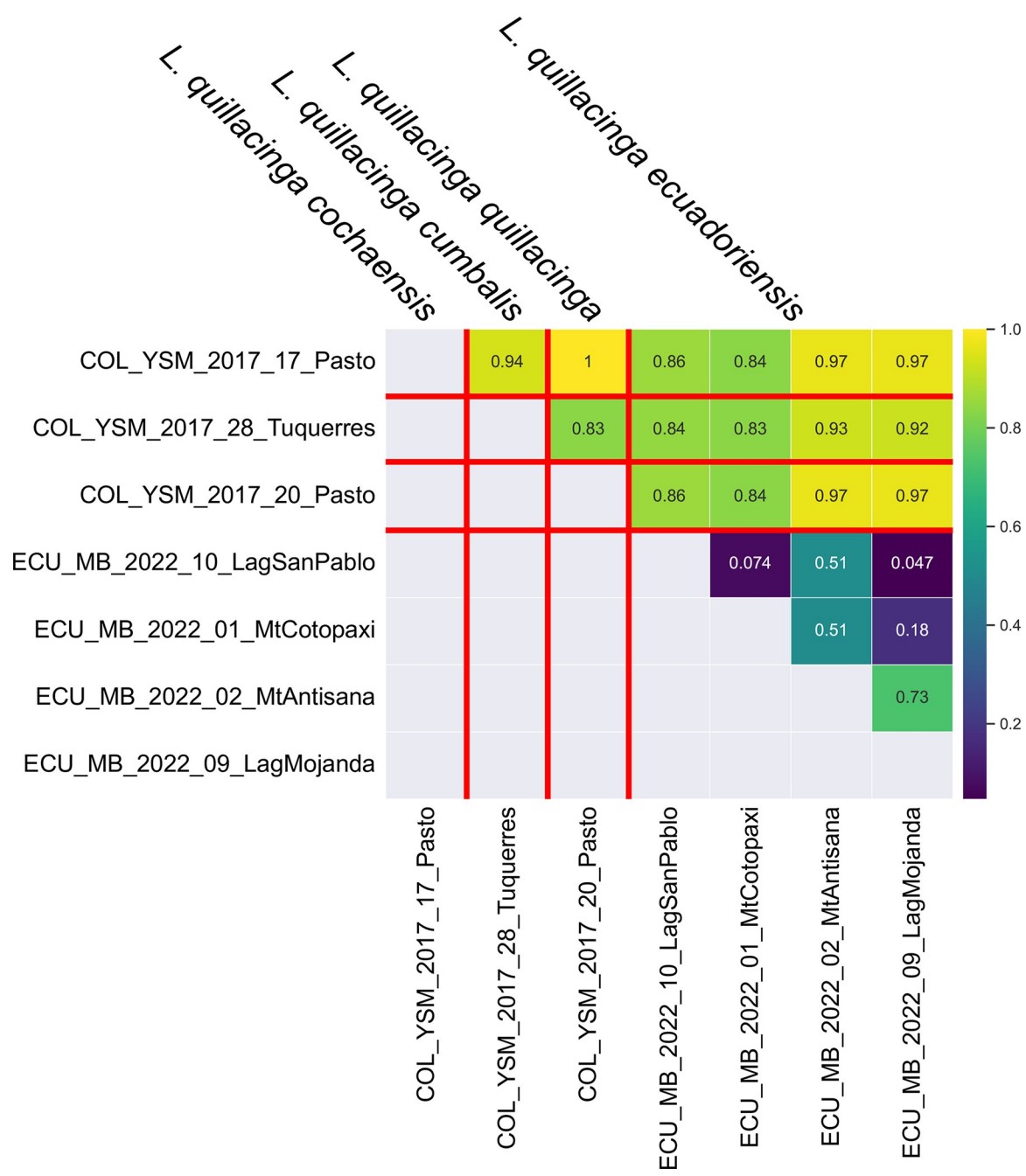

**Fig 7. Heatmap of pairwise comparison of Fst values among *L. quillacinga* sublineages.**

urn:lsid:zoobank.org:act:E86478D4-F4F9-4311-A59F-FAB6DA66BFE3

**Type locality.** Ecuador: Imbabura, Otavalo, Laguna San Pablo, 0.199˚ -78.2331˚. **Holotype.** Ecuador: Imbabura, Otavalo, Laguna San Pablo, humedal, 2,700 m, 28.xi.2022, 0.199˚ -78.2331˚, Basantes & Balke (ECU_MB_2022_10) (INABIO). **Paratypes.** 334 specimens with same data as the holotype (INABIO, ZSM); 75 exs., Ecuador: Pichincha, Cotopaxi, small wet patch 3,700 m, 21.xi.2022, -0.5657˚ -78.4438˚, Basantes & Balke (ECU_MB_2022_01) (ZSM); 246 exs., Ecuador: Napo, Antisana, La Mica area, 3,950 m, 22.xi.2022, -0.5361˚ 78.231˚,

**Table 1. Nucleotide diagnostic characters in the COI alignment for sublineages of *Liodessus quillacinga* (above) and *Liodessus quimbaya* (below).**

| !Taxon, Number of sequences | 28 | 88 | 202 | 316 | 322 | 325 | 343 | 433 | 628 |
|---|---|---|---|---|---|---|---|---|---|
| *L. quillacinga quillacinga* (5) | | C | | G | | | | | |
| *L. quillacinga cumbalis* (5) | | C | C | G | | | | G | |
| *L. quillacinga cochaensis* (5) | | | | | | A | T | | |
| *L. quillacinga ecuadoriensis* (44) | G | | | | G | | | | A/G |

| Taxon, Number of sequences | 40 | 55 | 124 | 178 | 229 | 259 | 271 | 283 | 298 | 325 | 440 | 472 | 494 | 592 |
|---|---|---|---|---|---|---|---|---|---|---|---|---|---|---|
| *L. quimbaya quimbaya* (24) | G | A | | | | | | | | | | | | |
| *L. quimbaya azufralis* (106) | | | G | | C | G | A | | C | A(G) | T | | T | |
| *L. quimbaya paletara* (20) | | | G | | | | | G | | | | G | | G(A) |

Basantes & Balke (ECU_MB_2022_02) (INABIO, ZSM); 165 exs., Ecuador: Imbabura, Laguna Mojanda, wet plant carpet, 3,750 m, 27.xi.2022, 0.1338˚ -78.2598˚, Basantes & Balke (ECU_MB_2022_09) (INABIO, ZSM).

**Size of holotype.** Total length 2.0 mm; length without head 1.8 mm; maximum width 1.0 mm. **Variation.** ($N$ = 41). Total length 1.8–2.3 mm; length without head 1.5–2.0 mm; maximum width 0.9–1.1 mm.

**Metathoracic wings.** Fully developed.

**Male genitalia.** Median lobe of aedeagus ventrally slim. In lateral view, simply curved parallel-sided, without broadened medially. In ventral view, lateral margins parallel basally, and in the median third conically narrowed towards the tip (Figs 4D–5G, 5D–5H).

**Diagnosis.** This new subspecies is very similar to other populations and can only be reliably separated on the basis of the locality and 5-COI data. Its aedeagus is ventrally and laterally slimmer and more parallel-sided. Ventrally, the median third is more distinct conically narrowed towards the tip. Laterally, it is not medially broadened than in *L. quillacinga quallacinga* and its two other subspecies (Figs 4 and 5). The coloration of the elytra shows a high variability (Figs 3D–5F, 8A–8D and 9). *Liodessus quillacinga ecuadoriensis* ssp. nov. is characterized geographically, known from several localities in northern Ecuador. In the COLLI sequence database, this new subspecies has three diagnostic characters different from the other sublineages (Table 1). The haplotype network structure is depicted in Fig 6

**Distribution.** *Liodessus quillacinga ecuadoriensis* ssp. nov. is distributed at altitudes between 2,700 and 3,950 m in northern Ecuador (Fig 2).

**Habitat.** Collected from very shallow water with dense vegetation, even small saucer sized water accumulations around lagoons (Fig 10A–10C).

**Etymology.** With the choice of the epithet "*ecuadoriensis*", the new subspecies is named after the country of Ecuador, which includes the type localities. The name is an adjective in the nominative singular.

***Liodessus quimbaya quimbaya* Megna, Hendrich & Balke, 2019.** Figs 11, 12, 13A, 14E, 15B, 16A, 16D, 17 and 19

**Type locality.** Colombia: Quindío, Salento, Paramo Frontino, 4.608˚ -75.428˚. **Holotype.** Colombia, Quindío, Salento, Paramo Frontino, 3,700 m, 25.iv.2017, 4.608˚ -75.428˚, Megna, Osorio & Vivero (CO-14) (LIAUN). **Paratypes.** See Megna et al. [16].

**Additional material.** 23 exs., Colombia, Cauca, Malvaza, 3,500 m, 29.i.2019, 2.483˚ -76.233˚, Prieto (COL_CPr_2019_01) (UNAL, ZSM).

**Description of holotype.** Total length 2.0 mm; length without head 1.8 mm; maximum width 1.0 mm. Shape elongate, with lateral outline only slightly discontinuous between pronotum and elytron. Pronotum broadest in its midlength, sides not very convex. Elytra widest at about midlength.

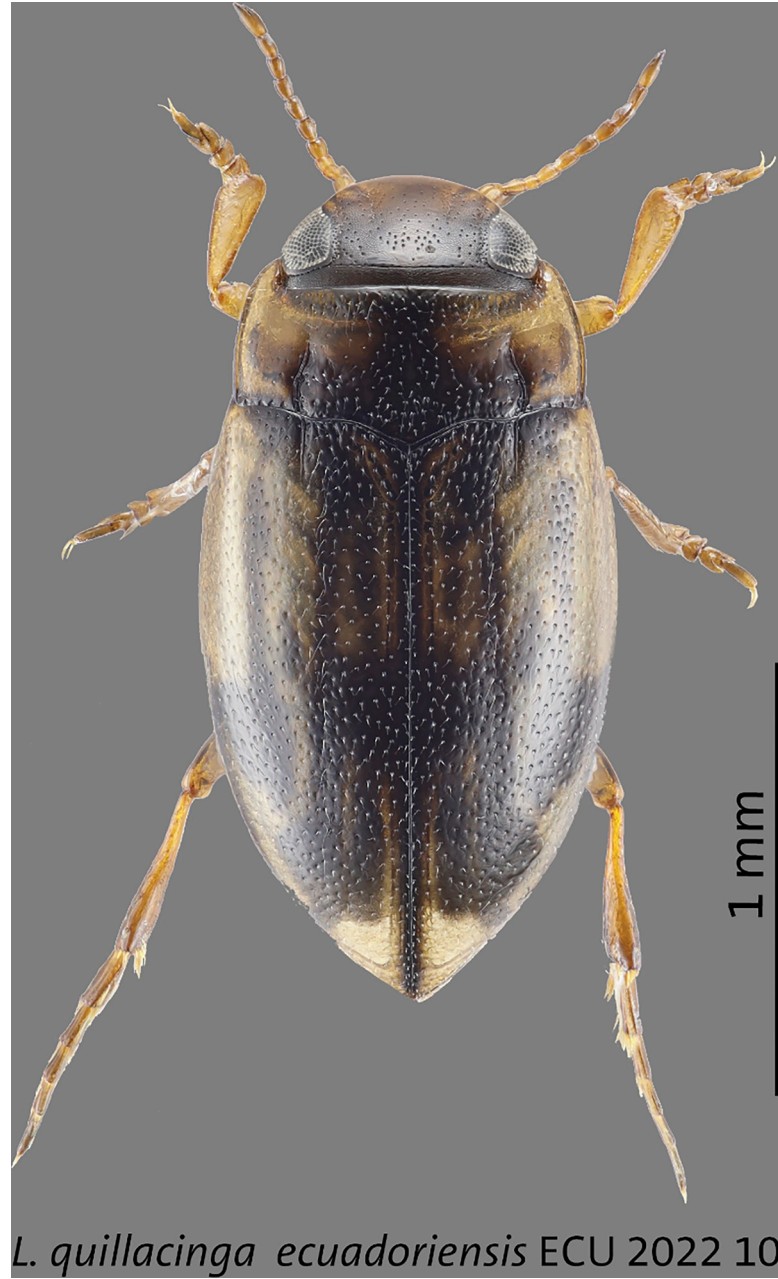

**Fig 8. Holotype of *Liodessus quillacinga ecuadoriensis* ssp. nov. from the Laguna San Pablo, 2,700 m, 0.199˚ -78.2331˚.** Scale bar: 1 mm.

**Colour.**   Dorsal side and ventrally dark brown to black. Antenna, mouthparts, pro-, and mesothoracic legs testaceous. Metathoracic leg and antennomeres largely infuscate.

**Surface sculpture.**   Head with distinct microreticulation and few punctures. Pronotum mostly shiny, anteriorly with faint microreticulation; with unevenly distributed coarse setiferous punctation. Elytron shiny; surface with dense and coarse setiferous punctation.

**Structures.**   Head with distinct occipital line. Pronotum with lateral bead; with distinct and very deep basal striae. Elytron with short and hardly defined basal striae.

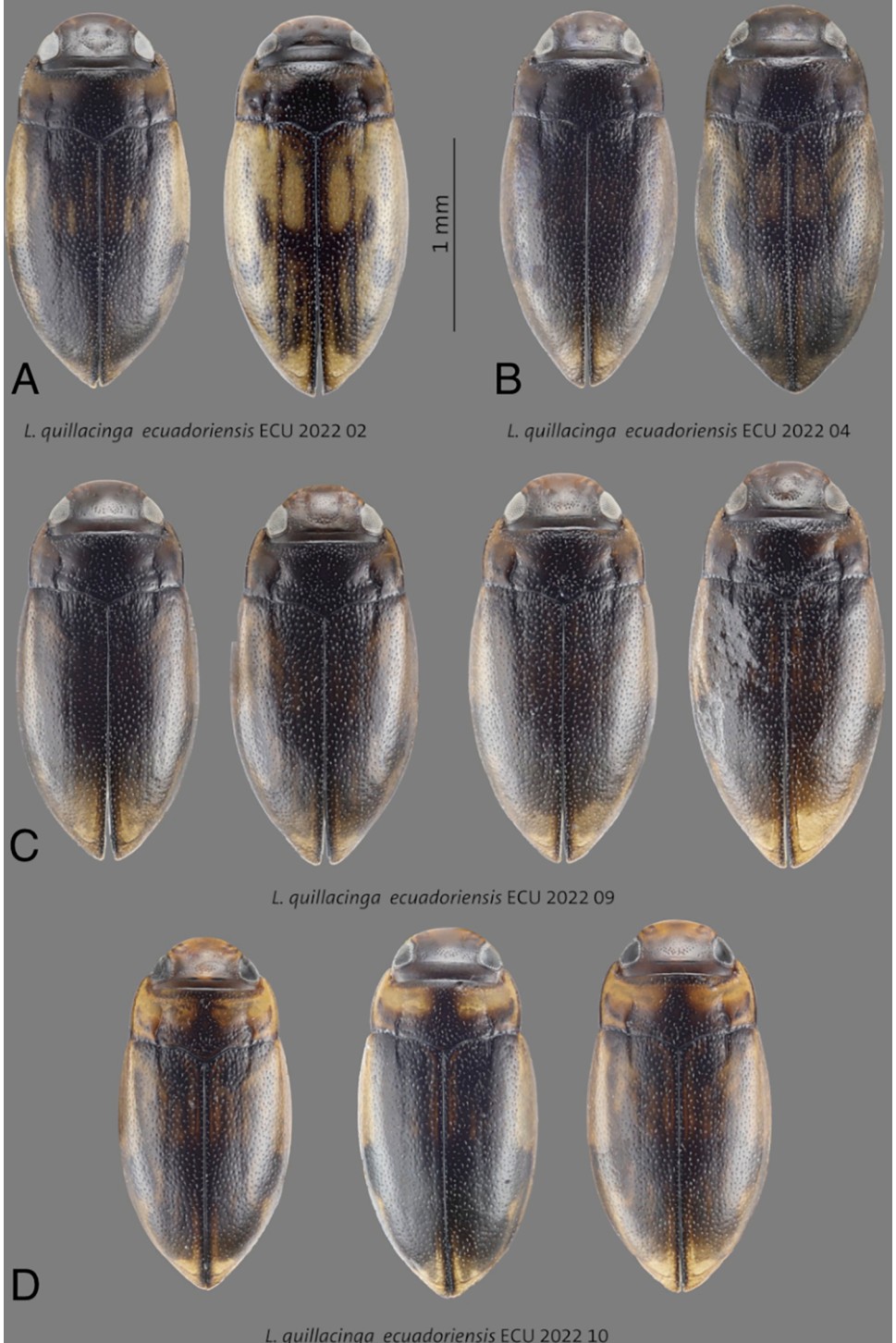

**Fig 9. A–D** Habitus of *Liodessus quillacinga ecuadoriensis* ssp. nov. paratypes from different localities.

**Genitalia.** Median lobe of aedeagus in lateral view longish, simply curved, apically narrowed, in ventral view lateral margins slightly narrowed medially and tip narrowed.

**Metathoracic wings.** Vestigial, reduced to short membranous stubs without sign of venation.

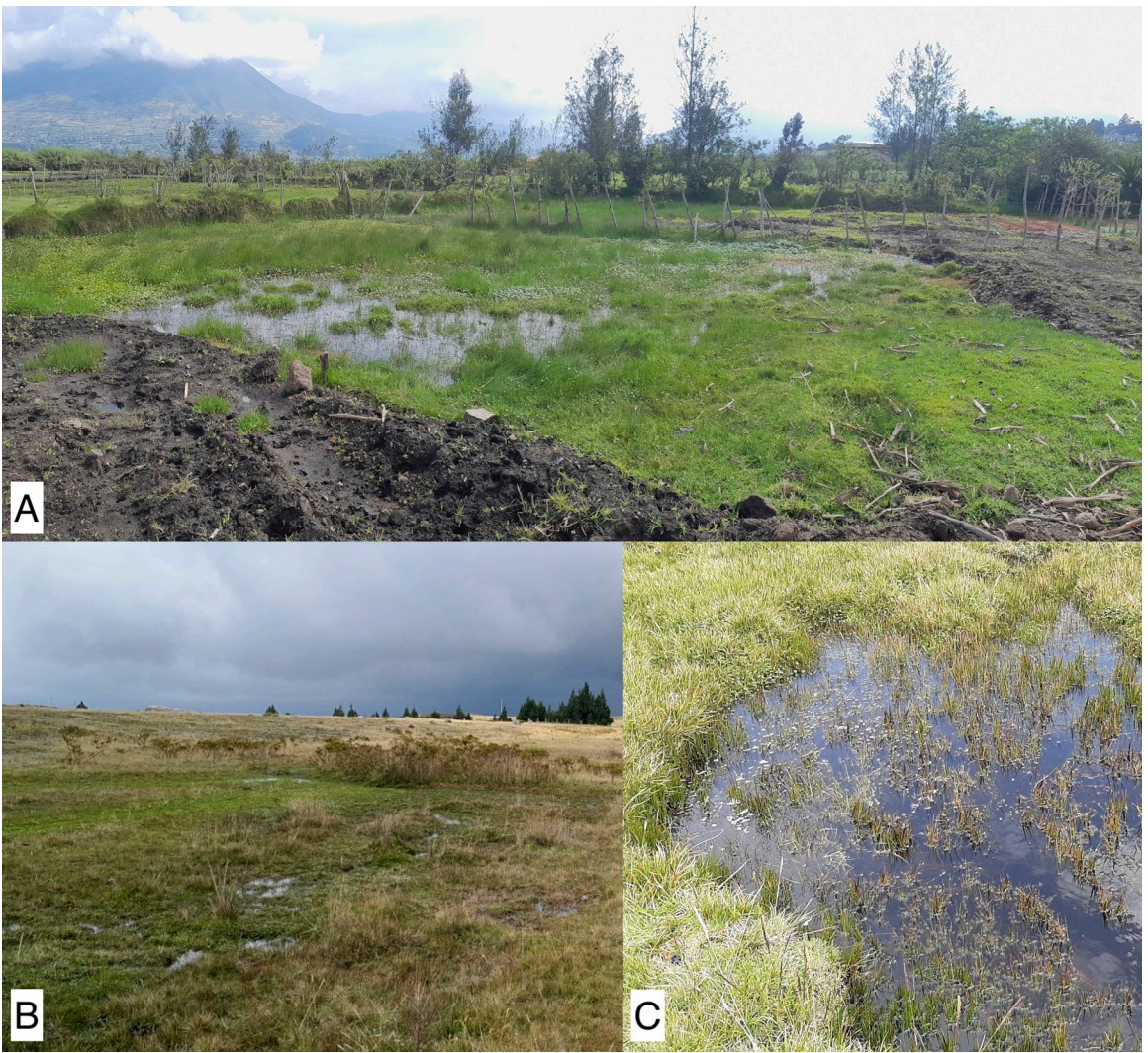

**Fig 10.** Habitats of *Liodessus quillacinga ecuadoriensis* ssp. nov., **A** Otavalo, Laguna San Pablo (locus typicus); **B** Mt. Cotopaxi, small wet patch; **C** Mt. Antisana, La Mica area.

**Variation.** Total length 1.9–2.2 mm (*N* = 22); length without head 1.6–1.9 mm; maximum width 0.9–1.0 mm. The extent of the basal elytral striae varies from absent to fairly well expressed. The dorsal coloration can be brighter, dark orange, laterally on pronotum and elytron

in some specimens. The structure of median lobe of the aedeagus in lateral and ventral view is constant in general shape but the curvature varies as illustrated in Figs 14E, 15B, 16A, 16D.

**Female.** Dorsal side dull, with strong microreticulation.

**Identification notes.** In the COLLI sequence database, *L. quimbaya quimbaya* has 2 diagnostic characters different from the other sublineages (Table 1). The haplotype network structure is depicted in Fig 18.

**Distribution.** To date only known from the Central Cordillera in southern Colombia at altitudes between 3,290 and 3,700 m (Fig 11).

**Habitat.** Páramo water body with compact aquatic vegetation, Fig 11a in Megna et al. [16].

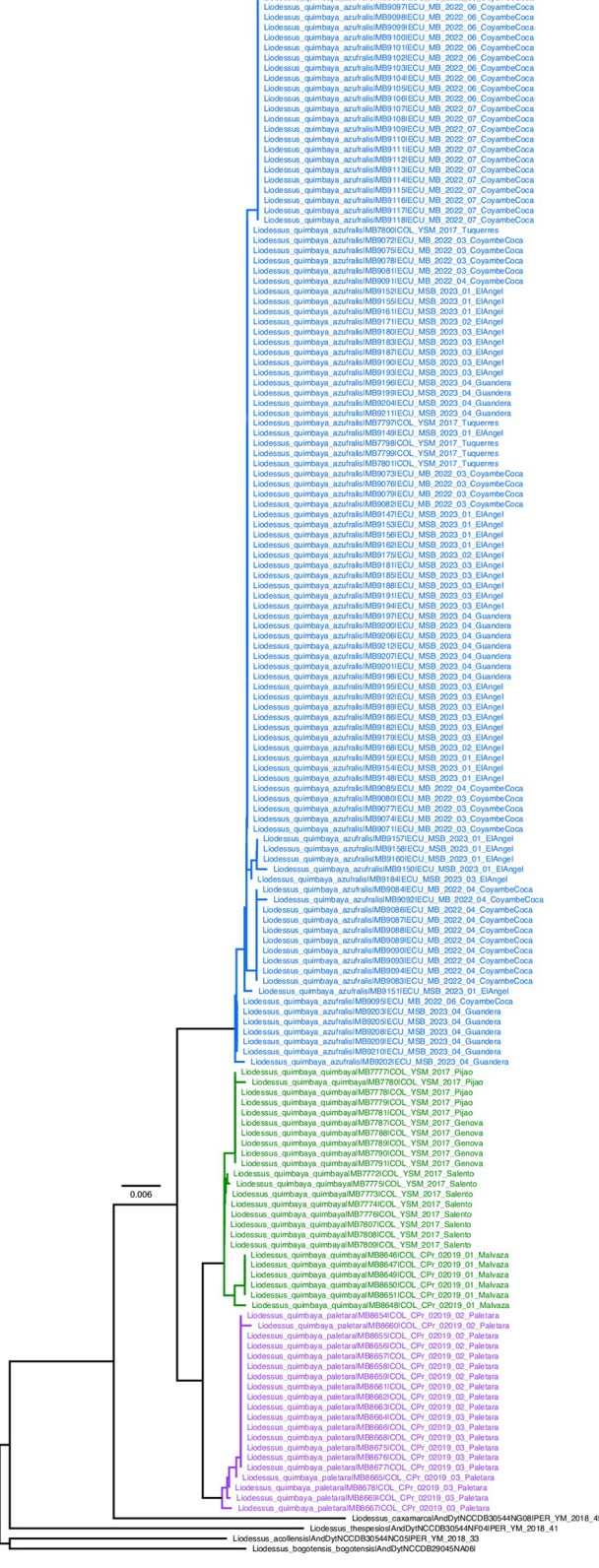

**Fig 11. Neighbour-joining tree for the sublineages of *Liodessus quimbaya*.** Blue *L. quimbaya azufralis* stat. nov., **Green** *L. quimbaya quimbaya*, **Pink** *L. quimbaya paletara* ssp. nov., **Black** outgroup.

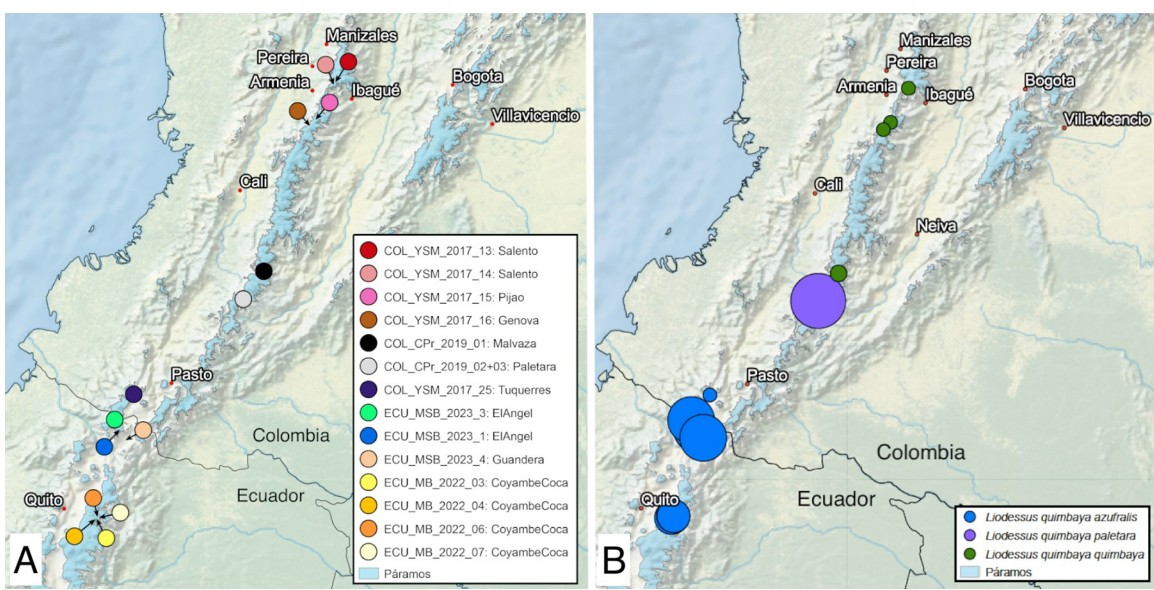

**Fig 12.** Maps of **A** sampling localities and **B** lineages of *Liodessus quimbaya*. Blue coloured areas represent Páramo habitats.

***Liodessus quimbaya azufralis* Megna, Hendrich & Balke, 2019 stat. nov.** Figs 11, 12, 13A, 13C, 13D, 14C, 14D, 15A–15D, 17B, 17E–17I, 18 and 19

**Type locality.** Colombia, Narino, Surroundings of the Azufral volcano, 1.087˚, -77.711˚. **Holotype.** Colombia, Narino, Municipality of Túquerres, Surroundings of the Azufral volcano, 3,795 m, 07.v.2017, 1.087˚, -77.711˚, Megna & Ruiz (CO-25) (LIAUN). **Paratypes.** See Megna et al. [16].

**Additional material.** 197 exs., Ecuador: Napo, Coyambe Coca, puddle, 3,700 m, 23. xi.2022, -0.3169˚ -78.1423˚, Basantes & Balke (ECU_MB_2022_03) (INABIO); 182 exs., "Ecuador: Napo, Coyambe Coca, small round lagoon, 3,900 m, 24.xi.2022, -0.3223˚ -78.1583˚, Basantes & Balke (ECU_MB_2022_04) (INABIO, ZSM); 141 exs., "Ecuador: Napo, PN Coyambe Coca, small lagoon, 4,000 m, 26.xi.2022, -0.3007˚ -78.1338˚, Basantes & Balke (ECU_MB_2022_06) (INABIO, ZSM); 111 exs., Ecuador: Napo, PN Coyambe Coca, puddle, 3,950 m, 26.xi.2022, -0.3018˚ -78.1303˚, Basantes & Balke (ECU_MB_2022_07) (INABIO, ZSM); 210 exs., Ecuador: Carchi, Ecological Reserve El Ángel, 3,750 m, 14.ii.2023, 0.6854˚ -77.8766˚, Basantes (ECU_MSB_2023_01) (INABIO, ZSM); 329 exs., Ecuador: Carchi, Ecological Reserve El Ángel (Lagunas verdes), 3,950 m, 15.ii.2023, 0.7991˚ -77.9279˚, Basantes (ECU_MSB_2023_03) (INABIO, ZSM); 108 exs., Ecuador: Carchi, Biological Station Guandera (nearby puddles), 2,750 m, 16.ii.2023, 0.59522˚ -77.78842˚, Basantes (ECU_MSB_2023_04) (INABIO, ZSM).

**Note.** Megna, Hendrich & Balke (in Megna et al. [16]) described *L. quimbaya* and *L. azufralis* at species level. In particular, morphological analysis of new material from further localities indicates a much greater similarity of the two taxa than assumed in the original description. Therefore, it is suggested that both taxa belong to one species, with one subspecies and names made available in the same paper (Art. 8, Art. 10., Art. 11, Art. 16, Art. 23, ICZN [32]). As *L. quimbaya* was described on pages 10ff. and *L. azufralis* on pages 14ff. of Megna et al. [16], *L. quimbaya* is chosen as priority name for this taxon, acting as "First Revising Author" according to Art. 24.2 of ICZN [32]. Accordingly, the new combination must be *L. quimbaya azufralis* stat. nov. for this taxon.

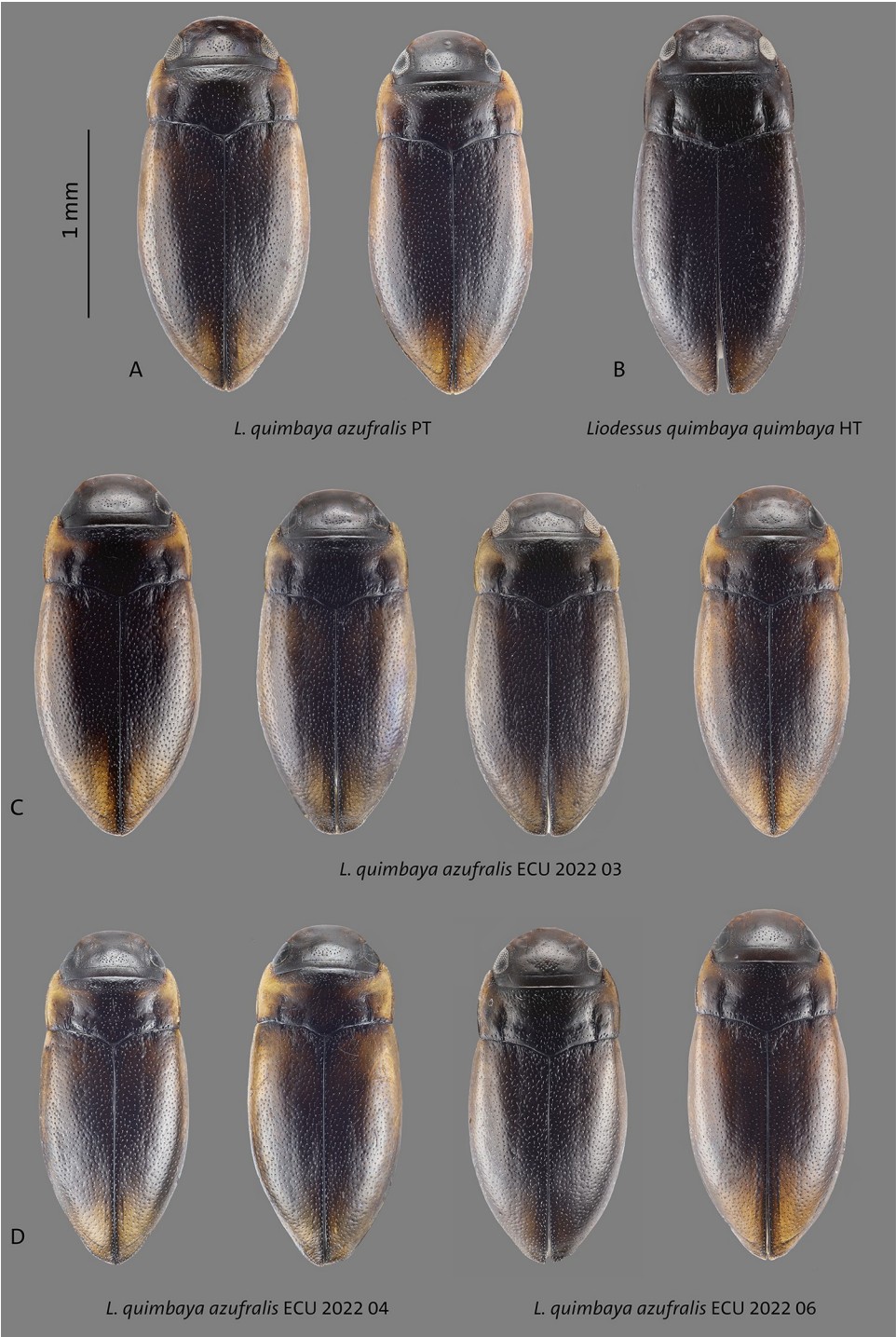

**Fig 13.** Habitus of **A** *L. quimbaya azufralis* stat. nov. paratypes (PT), **B** *L. quimbaya quimbaya* holotype (HT), **C–D** *L. quimbaya azufralis* stat. nov. from different localities. Scale bar: 1 mm.

**Size of holotype.** Total length 2.1 mm; length without head 1.8 mm; maximum width 1.0 mm. **Variation.** Total length 1.7–2.1 mm ($N$ = 60); length without head 1.5–1.9 mm; maximum width 0.8–1.0 mm.

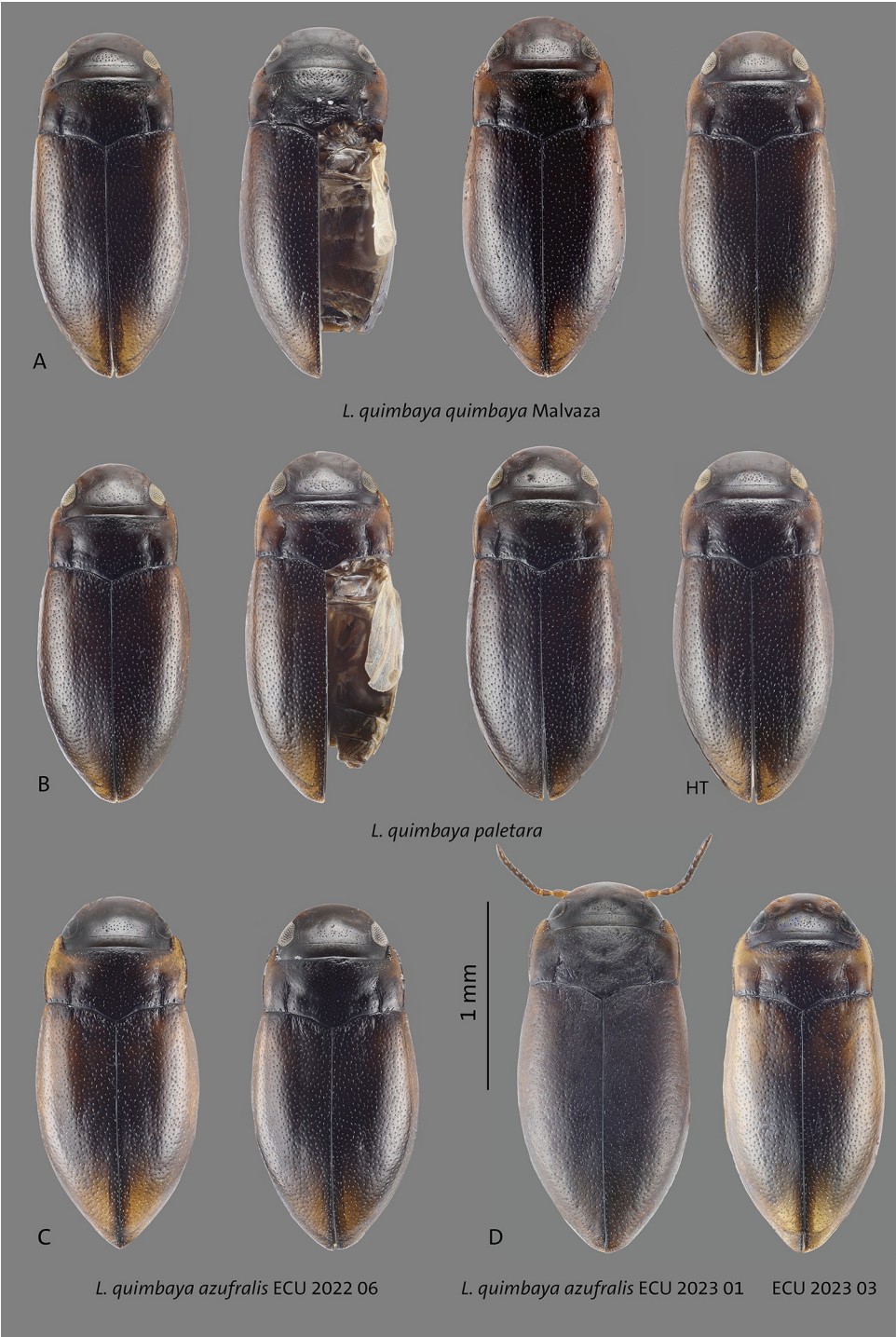

**Fig 14.** Habitus of **A** *L. quimbaya quimbaya* from Malvaza, **B** *L. quimbaya paletara* ssp. nov. paratypes and holotype (HT), **C–D** *L. quimbaya azufralis* stat. nov. from different localities. Scale bar: 1 mm.

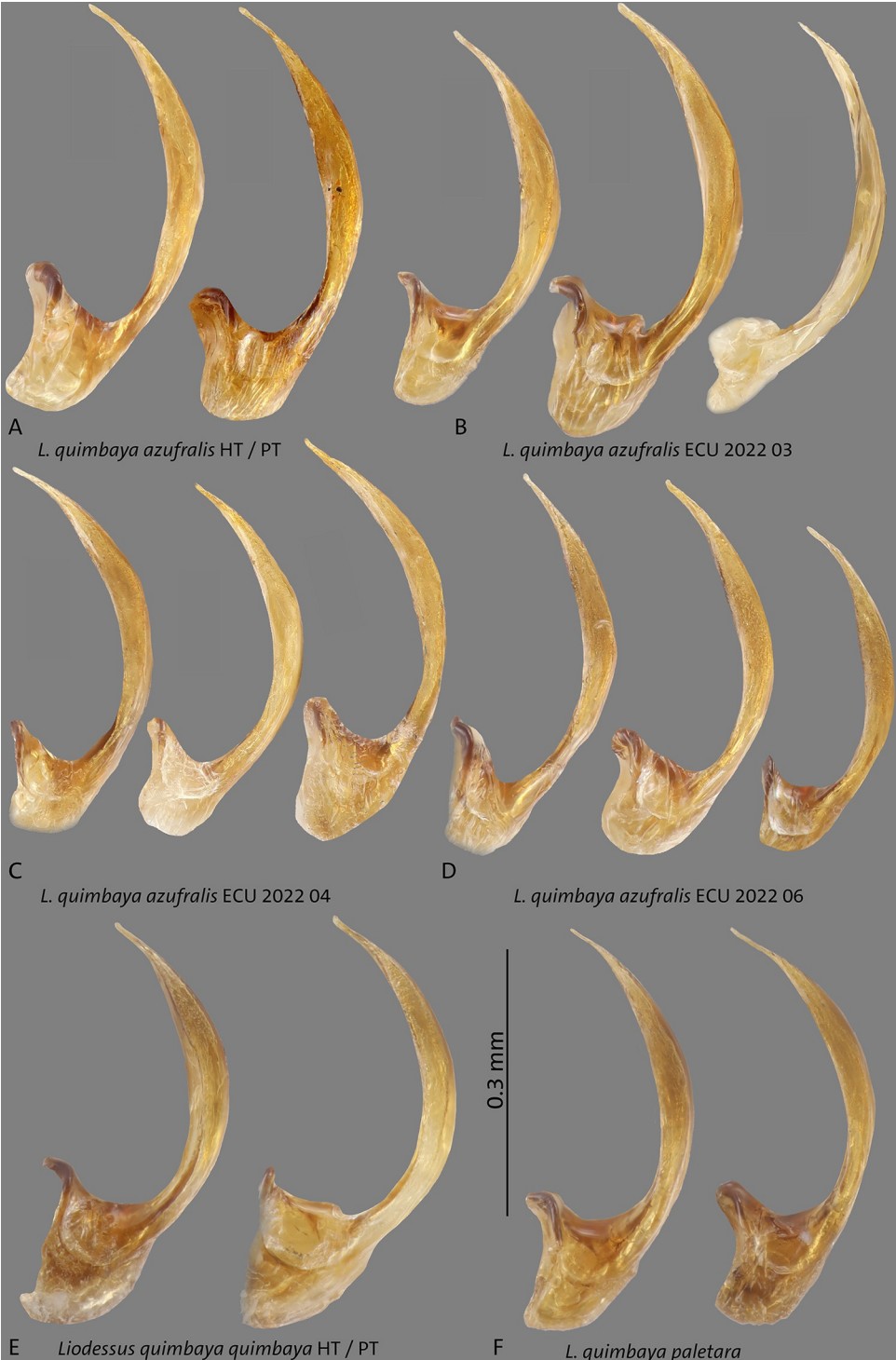

**Fig 15.** Median lobes in lateral view of **A** *L. quimbaya azufralis* stat. nov. holotype (HT) and paratype (PT), **B–D** from Ecuador; **E** *L. quimbaya quimbaya* holotype (HT) and paratype (PT), **F** *L. quimbaya paletara* ssp. nov. Scale bar: 0.3 mm.

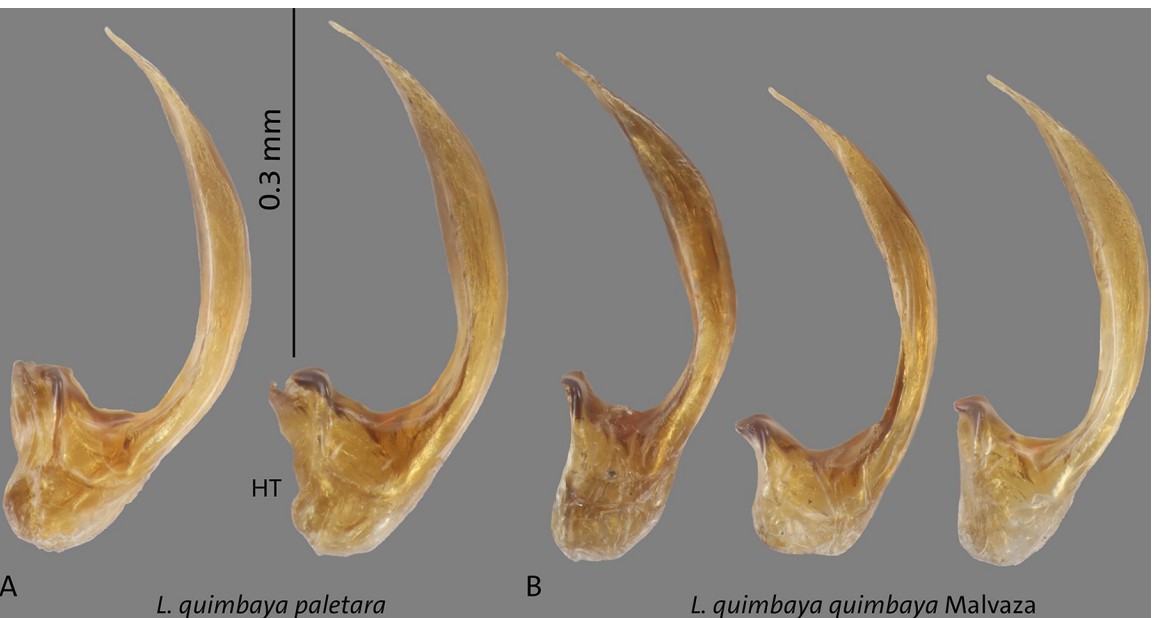

**Fig 16.** Median lobes in lateral view of *Liodessus* species **A** *L. quimbaya paletara* ssp. nov. paratype and holotype (HT), **B** *L. quimbaya quimbaya* from Malvaza. Scale bar: 0.3 mm.

**Metathoracic wings.** Vestigial, reduced to short membranous stubs without sign of venation in random specimens dissected from all populations.

**Diagnosis.** This subspecies is morphologically very similar to other populations. The habitus of *L. quimbaya azufralis* stat. nov. is however more broadly oval and drop shaped than in *L. quimbaya quimbaya* and *L. quimbaya paletara* ssp. nov. (Figs 13 and 14). The dorsal coloration is mostly brighter. This subspecies is characterized geographically, occurring in southern Colombia as well as northern Ecuador. In the COLLI sequence database, this subspecies has eight diagnostic characters different from the other sublineages (Table 1). The haplotype network structure is depicted in Fig 18.

**Distribution.** *Liodessus quimbaya azufralis* stat. nov. is widely distributed in the Pasto Massif of southern Colombia and northern Ecuador at altitudes between 2,700 and 4,000 m (Fig 12).

**Habitat.** Mostly collected from very shallow water with dense vegetation, even small saucer sized water accumulations around lagoons or in small depressions along the road; also collected from deeper water (up to 40 cm) at along the edge of a small, round lagoon, with green algae and vegetation at the edge and body of the lagoon.

***Liodessus quimbaya paletara* ssp. nov.** Figs 11, 12, 14B, 15F, 16A, 17C, 18 and 19
urn:lsid:zoobank.org:act:D9FB958C-1F95-4D23-ACC3-0F5BDB1439AD

**Type locality.** Colombia, Cauca, Paletará Valley, 2.167˚ -76.466˚. **Holotype.** Colombia, Cauca, Paletará, 2,850 m, 30.iv.2019, 2.167˚ -76.466˚, Prieto (COL_CPr_2019_03) (UNAL, ZSM). **Paratypes.** 134 exs., same label data (UNAL, ZSM); 82 exs., Colombia, Cauca, Paletará, 2,900 m, 11.ix.2019, 2.167˚ -76.466˚, Prieto (COL_CPr_2019_02) (UNAL, ZSM).

**Size of holotype.** Total length 1.9 mm; length without head 1.6 mm; maximum width 0.9 mm. **Variation.** ($N$ = 11) Total length 1.9–2.0 mm; length without head 1.6–1.7 mm; maximum width 0.9 mm.

**Metathoracic wings.** Vestigial, reduced to short membranous stubs without sign of venation in random dissected specimens from both localities.

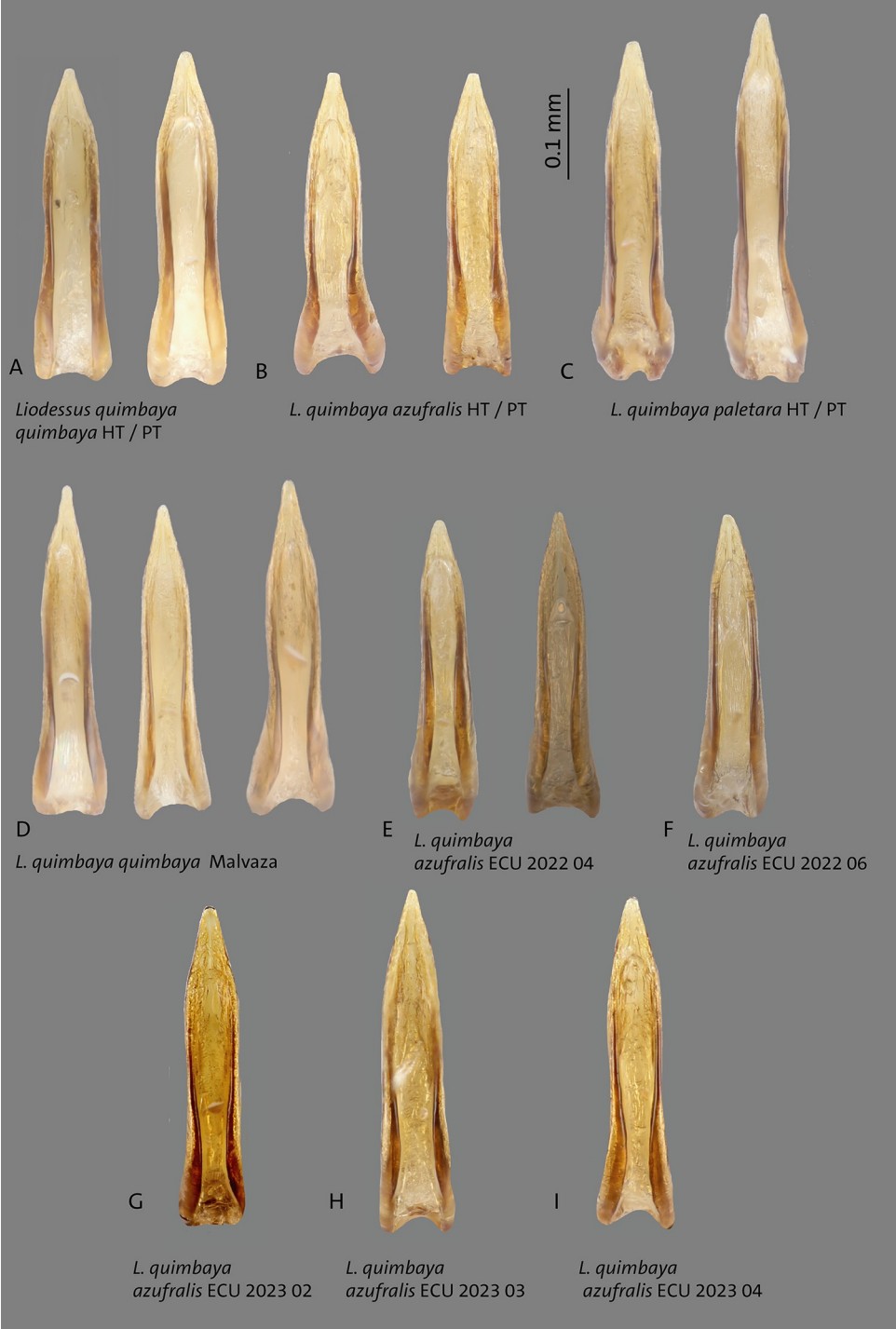

**Fig 17.** Median lobes in ventral view of *Liodessus* species **A** *L. quimbaya quimbaya* holotype (HT) and paratype (PT), **B** *L. quimbaya azufralis* stat. nov. holotype (HT) and paratype (PT), **C** *L. quimbaya paletara* ssp. nov. holotype (HT) and paratype (PT), **D** *L. quimbaya quimbaya* from Malavaza, **E–I** *L. quimbaya azufralis* stat. nov. from different localities. Scale bar: 0.1 mm.

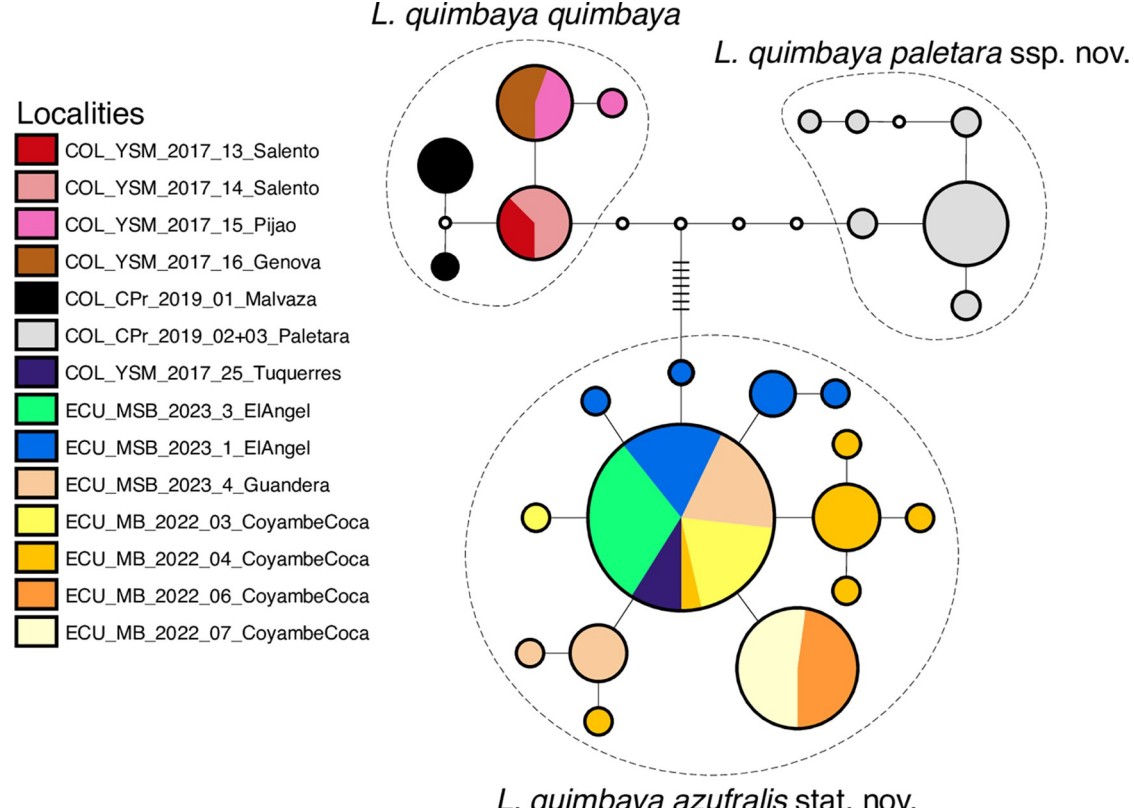

**Fig 18. Haplotype network for *Liodessus quimbaya*, including information of main sublineages.**

**Diagnosis.** This new subspecies is very similar to other populations. It is well-characterized geographically. In the COLLI sequence database, this new subspecies has four diagnostic characters different from the other sublineages (Table 1). The haplotype network structure is depicted in Fig 18.

**Distribution.** To date only known from the Paletará Valley in the Central Cordillera of southern Colombia at altitudes between 2,850 and 2,900 m (Fig 12).

**Habitat.** Collected from very shallow water with dense vegetation next to the road from Paletará to San José de Isnos.

**Etymology.** With the choice of the epithet "*paletara*", this new subspecies is named after its type locality, the Paletará Valley in the Colombian Massif. The name is a noun in the nominative singular standing in apposition.

## Notes on *Liodessus riveti* Peschet, 1923

*Liodessus riveti* Peschet, 1923 was described from Ecuador, data: El Pelado, 4,150 m, January 1903, leg. P. Rivet. The holotype is deposited in the Museum National d'Histoire Naturelle in Paris [19], but could not be located during a stay of M. Balke in 2017. Without re-examination and only based on the original description, no reliable taxonomic assessment can be made.

In the World Catalogue of Dytiscidae [11], the type locality is given as El Pelado (island), referring to an islet in the El Pelado marine reserve of Ecuador. This is most likely incorrect, as the altitude given by Peschet [19] indicates an alpine area. Paul Rivet did in fact explore the high-altitude regions of Ecuador, and in January 1903, he worked in the area of Tulcán in

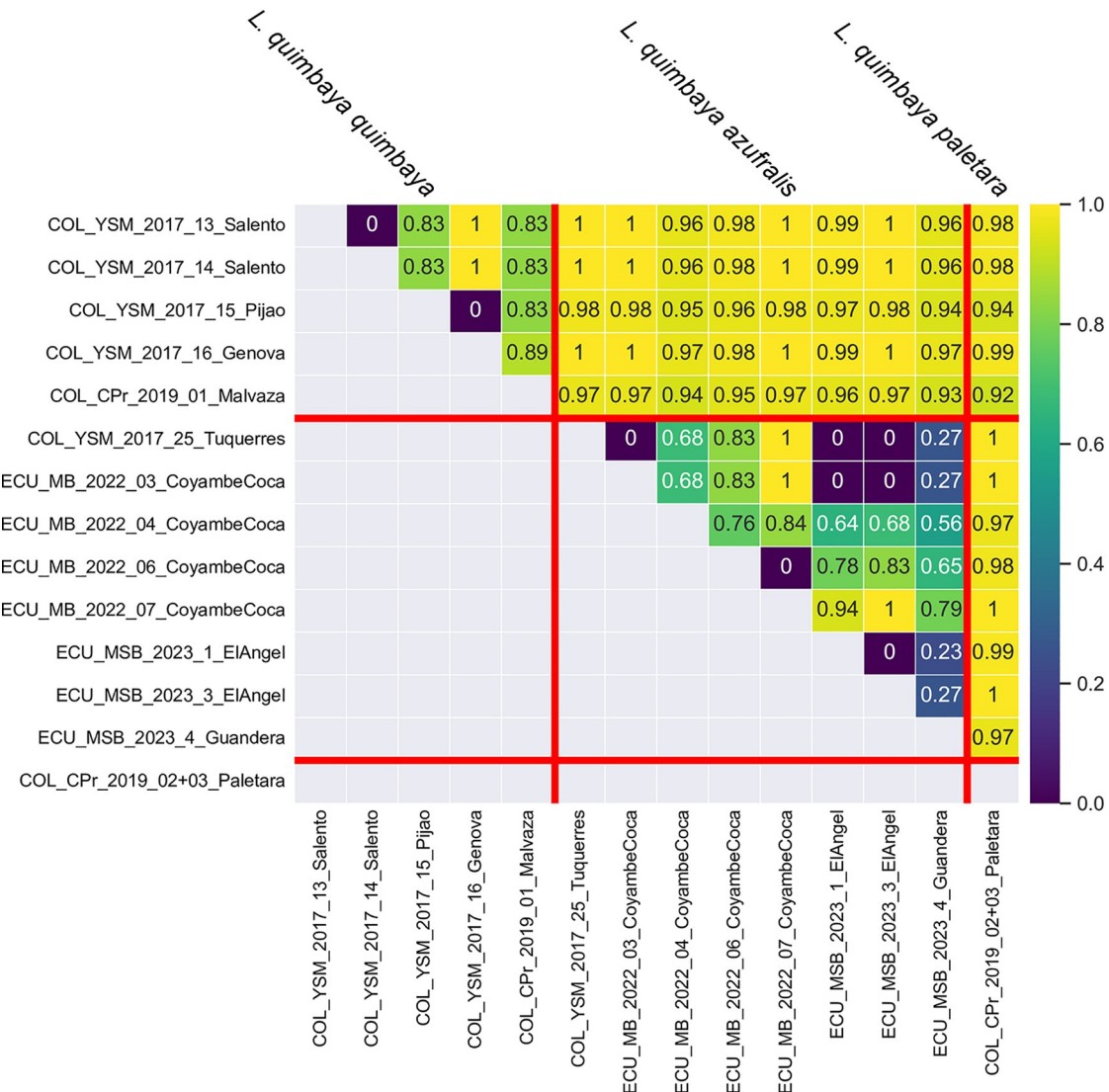

**Fig 19. Heatmap of pairwise comparison of Fst values among *L. quimbaya* sublineages.**

northern Ecuador [33]. It seems adequate to infer that El Pelado refers to the Cerro El Pelado, part of the Reserva Ecológica El Ángel, situated at 0.7268˚ -77.9219˚.

The team of Michael Balke has made extensive collections in that reserve recently. The *Liodessus* collected there were all assigned to *L. quimbaya azufralis* stat. nov. (see above), so that re-examination of the type of *L. riveti* might reveal the conspecificity of both taxa. This would make *L. quimbaya* a younger synonym of *L. riveti* and lead to the corresponding renaming of its herein discussed sublineages (Art. 23, ICZN [32]).

## Molecular evidence

The complete DNA alignments comprise DNA sequences of 147 specimens of *Liodessus quimbaya* and 59 specimens of *Liodessus quillacinga*, for which 124 and 44 are newly sequenced respectively. Neighbour joining trees (Figs 1 and 11) and haplotype networks (Figs 6 and 19) provide complementary and compatible data, clustering sequences/haplotypes in various

molecular operational taxonomical units that we have considered as valid subspecies (in agreement with the morphological and geographical evidence). The species delimitation methods we applied yield contrasting results (S1 and S2 Figs): while both methods performed similarly for *Liodessus quimbaya*, providing 2–3 clusters consistent with the other methodologies (S1 Figs), their performance for *Liodessus quillacinga* was inconsistent, producing from 3 to 16 clusters (S2 Figs). Therefore, we opt to not rely on these approaches for taxonomical decisions.

**Liodessus quillacinga.** The neighbour-joining tree (Fig 1) breaks down the samples into four distinct groups, in agreement with the four geographically structured clusters identified in the haplotype network (Fig 6). The three *L. quillacinga* subspecies, as defined by Megna et al. [16], are each represented by isolated localities in southern Colombia, whereas the newly described *L. quillacinga ecuadoriensis* ssp. nov. corresponds to a wider geographic sampling from northern Ecuador. Pairwise comparison of *L. quillacinga* localities (Fig 7) underscores substantial isolation and differentiation for each subspecies (Fst ≥ 0.83). In contrast, *L. quillacinga ecuadoriensis* ssp. nov. exhibits robust intrapopulation connectivity (Fst < 0.73), with the majority of comparisons below 0.51, indicating a comparatively lower level of differentiation and a higher level of gene flow between populations.

**Liodessus quimbaya.** In the case of *L. quimbaya*, the neighbour-joining tree shows a distinct clustering of localities into three geographically structured groups (Fig 11). Specifically, samples from Ecuador and the Colombian locality near the Ecuadorian border (Túquerres) group together. This group (*L. quimbaya azufralis* stat. nov.) shows a robust intrapopulation connectivity (Fst ≤ 0.84). Samples collected from Paletará (Colombia) constitute a separate clade (*L. quimbaya paletara* ssp. nov.). The remaining samples, situated in the northern region of the sampling area, form the third distinct group. As can be seen in the haplotype network (Fig 18), these three clusters match the three subspecies identified in this study. The connectivity pattern exhibited by the heatmap (Fig 19) is not as clear as is in *L. quillacinga*, yet it reveals some level of isolation and differentiation at both intra- and interpopulation levels. Nonetheless, it highlights a comparatively lower genetic differentiation (indicated as colder colours in the heatmap) within each subspecies.

## Updated checklist of *Liodessus* from Colombia and Ecuador

**(* taxa treated herein).** *Liodessus bogotensis bogotensis* Guignot, 1953 (Colombia)

*Liodessus b. almorzadero* Balke et al., 2023 (Colombia)

*Liodessus b. chingaza* Balke et al., 2023 (Colombia)

*Liodessus b. lacuniviridis* Balke et al., 2020 (Colombia)

*Liodessus b. matarredonda* Balke et al., 2023 (Colombia)

*Liodessus b. sumapaz* Balke et al., 2023 (Colombia)

*Liodessus picinus* Balke et al., 2021 (Colombia)

*Liodessus quillacinga quillacinga* Megna, Hendrich & Balke, 2019 (Colombia) *

*Liodessus quillacinga cochaensis* Megna, Hendrich & Balke, 2019 (Colombia) *

*Liodessus quillacinga cumbalis* Megna, Hendrich & Balke, 2019 (Colombia) *

*Liodessus quillacinga ecuadoriensis* ssp. nov. (Ecuador) *

*Liodessus quimbaya quimbaya* Megna, Hendrich & Balke, 2019 (Colombia) *

*Liodessus quimbaya azufralis* Megna, Hendrich & Balke, 2019 stat. nov. (Colombia, Ecuador) *

*Liodessus quimbaya paletara* ssp. nov. (Colombia) *

*Liodessus riveti* Peschet, 1923 (Ecuador) *

*Liodessus santarosita* Balke et al., 2023 (Colombia)

## Discussion

The integrative approach employed here has identified taxonomically informative characteristics, at the same time emphasizing those susceptible to pronounced variability. We also find that the number of putative species suggested by two different species delineation algorithms is highly inconsistent, which is however not surprising for probably very recently diverged populations [34]. Notably, we here find a greater intraspecific variability in the studied taxa than previously reported by Megna et al. [16], particularly concerning coloration and the extent of basal striae on pronotum and elytra. Molecular phylogenetic evidence revealed morphologically cryptic, geographically structured population-level diversity. The observed patterns find a potential explanation in the "flickering connectivity system" model of the Páramo ecosystem, outlined by Flantua et al. [2]. Shaped by the dynamic shifts between glacial and interglacial Pleistocene periods, the elevations at which Páramos developed underwent continuous changes over the last million years. This dynamism has fostered connections and disconnections among Páramos, influencing dispersal, gene flow and local extinction of Andean *Liodessus* species within Páramo ecosystems. This phenomenon may also account for recent speciation processes detected in the area, as well as the observed intraspecific variability and hybridization within the *Liodessus bogotensis* species complex–a related species from the eastern Cordillera of Colombia [10]. Notably, the "flickering connectivity system" plays a crucial role in shaping biodiversity across diverse taxa. This is underscored by its substantial influence on the present diversity of plants in the Páramos (ca. 5,000 species [5]) or by the recent diversification of the majority of bird species in the Páramo [35], which emerged after the late Miocene/early Pliocene. In addition, differences in β-diversity were found to divide the Páramos of the northern Andes (Colombia) and the southern Andes (Ecuador) into two spatially structurally distinct groups, which is explained by past climatic conditions [35].

A fascinating pattern of disconnection emerges from the detailed study of what we classify as *Liodesus quimbaya* subspecies: while each subspecies exhibits moderate intralineage variability, their interlineage Fst values indicate a significant disconnection. Particularly intriguing is the proximity of *L. quimbaya quimbaya*, covering a known distribution range of over 250 km from north to south, to *L. quimbaya paletara* ssp. nov., separated by a mere 40 km at the southern limit. The presence of the active Puracé volcano in the distribution boundary of both subspecies may have intensified their isolation. The volcanic activity, as proposed by Finn et al. [36], emerges as a plausible environmental influence contributing to the observed genetic structure, likely wiping out some populations and generating local bottlenecks (i.e., reduction of the genetic pool) that left a lasting genetic imprint. However, this area presents a valuable opportunity for further research to evaluate the stability of the proposed *Liodessus* classification and explore the intraspecific boundaries of *Liodessus quimbaya*.

The present study significantly contributed to our understanding of the geographical range of *L. quimbaya azufralis* stat. nov. by expanding its known distribution to northern Ecuador. In contrast to *Liodessus quillacinga*, all subspecies of *Liodessus quimbaya* exhibit a notable reduction in metathoracic wing size. Surprisingly, despite the anticipated impact of varying wing development on the potential dispersal capabilities (vestigial wings for *L. quimbaya* and fully developed wings for *L. quillacinga*), the latter (fully winged one) displays a higher number of lineages in a smaller geographical span. Interestingly, the findings indicate that the distinct dispersal capabilities of the two species do not seem to be the primary factor influencing their diversity. While distance and dispersal capabilities are not the main factors, the overall higher connectivity of the Ecuadorian Páramos (sensu Flantua et al. [2]) supports the idea that the changing extend of Páramo over time helps to explain the genetic and slight morphological variations observed. In the case of *Liodessus quimbaya quimbaya*, there appears to be evident

connectivity within the Colombia Central Cordillera between the Páramo districts of Viejo Caldas-Tolima and Macizo Colombiano allowing recent range expansion.

The flickering system, although not rigid, displays varying degrees of stability along the Andes, with the southern Andes (Ecuador) showing a higher degree of connectivity compared to the northern Andes in Colombia [2], occasionally interrupted by rare instances of fragmentation. This is supported by our results, suggesting the flickering connectivity system is a strong force driving intralineage connectivity and interlineage isolation, consequently fostering recent evolutionary expansion. The star-shaped [36–38] pattern of the haplotype clusters in the southern part of the geographic ranges in the sublineages of *L. quillacinga* and *L. quimbaya* indicates recent range expansion into the northern Ecuador region. Processes caused by the flickering system or genetic differentiation over time due to large geographical distances, could (re)separate populations in the future, hindering dispersal and gene flow. This, in turn, might lead to the evolution of new endemic species, potentially contributing to the development of cryptic diversity that have not yet have developed significant morphological differences [39, 40]. However, recent studies indicate that the evolution of different lineages does not necessarily have to be accompanied by the evolution of significant morphological differences, especially for species inhabiting extreme habitats, such as high alpine areas [41, 42]. Therefore, it is possible that the lineages discussed here are in the process of speciation, as illustrated by the high number of haplotypes for *L. quillacinga ecuadoriensis* ssp. nov. and *L. quimbaya azufralis* stat. nov. The observed patterns can be interpreted as "cryptic diversity", but whether this diversity will persist in the ongoing evolutionary process remains unclear. The isolated lineages in Colombia, displaying cryptic diversity, do not currently indicate the evolution of significant morphological characters. It is possible that isolation and speciation are not yet complete and may never be, as long as anthropogenic climate change does not result in the absence of further dynamic changes between glacial and interglacial periods [43].

The recent range expansion and lineage idiosyncratic population structure of *Liodessus* diving beetles in the high Andes of Ecuador and Colombia seems to be closely linked to climatic processes. Further investigations, in particular with much finer tuned population genomic methods, could therefore not only lead to new taxonomic results, but also improve the understanding of the relationship between climate protection and species conservation—one of the greatest challenges of the 21st century.

## Supporting information

**S1 Fig. Maximum Likelihood phylogeny of *L. quimbaya* including two species delimitation method: Best two cluster configuration based on ASAP and PTP.**
(TIF)

**S2 Fig. Maximum Likelihood phylogeny of *L. quillacinga* including two species delimitation method: Best two cluster configuration based on ASAP and PTP.**
(TIF)

## Author Contributions

**Conceptualization:** Michael Balke.

**Data curation:** Michael Balke, Tobias Mainda, Katja Neven, Lars Hendrich, Adrián Villastrigo.

**Formal analysis:** Michael Balke, Tobias Mainda, Adrián Villastrigo.

**Funding acquisition:** Michael Balke.

**Investigation:** Michael Balke, Tobias Mainda, Michael Steven Basantes, Carlos Prieto, Adrián Villastrigo.

**Methodology:** Michael Balke, Tobias Mainda, Adrián Villastrigo.

**Project administration:** Michael Balke.

**Resources:** Michael Balke.

**Visualization:** Tobias Mainda, Adrián Villastrigo.

**Writing – original draft:** Michael Balke, Tobias Mainda, Adrián Villastrigo.

**Writing – review & editing:** Michael Balke, Tobias Mainda, Katja Neven, Lars Hendrich, Michael Steven Basantes, Carlos Prieto, Adrián Villastrigo.

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
