## [Decision Letter · Decision Letter 0]

18 Jun 2024

PONE-D-24-13211Recent range expansion and lineage idiosyncratic population structure of Liodessus diving beetles in the high Andes (Coleoptera: Dytiscidae, Bidessini)PLOS ONE

Dear Dr. Villastrigo,

Thank you for submitting your manuscript to PLOS ONE. After careful consideration, we feel that it has merit but does not fully meet PLOS ONE’s publication criteria as it currently stands. Therefore, we invite you to submit a revised version of the manuscript that addresses the points raised during the review process.

I am pleased to inform you that two Reviewers positively assessed your manuscript. One of the Reviewers pointed out several issues that should be considered during the revision of this study. I encourage you to consider these comments (please check also the information attached to the pdf file) and respond if you disagree with some of them. Additionally, I recommend to add to genetic analyses one or two methods (e.g. one based on genetic gap like ABGD and second on phylogenetic reconstruction e.g  PTP or GMYC) of species delimitation to statistically support species distinctiveness. It should be possible based on available barcode sequences. 

We look forward to receiving your revised manuscript.

Kind regards,

Łukasz Kajtoch, Ph.D.

Academic Editor

PLOS ONE

Journal Requirements:

When submitting your revision, we need you to address these additional requirements. 1. Please ensure that your manuscript meets PLOS ONE's style requirements, including those for file naming. The PLOS ONE style templates can be found at https://journals.plos.org/plosone/s/file?id=wjVg/PLOSOne_formatting_sample_main_body.pdf and https://journals.plos.org/plosone/s/file?id=ba62/PLOSOne_formatting_sample_title_authors_affiliations.pdf 2. In your Methods section, please provide additional information regarding the permits you obtained for the work. Please ensure you have included the full name of the authority that approved the field site access and, if no permits were required, a brief statement explaining why. 3. Please take this opportunity to be sure you have met all of our guidelines for new species. For proper registration of a new zoological taxon, we require two specific statements to be included in your manuscript.A.
In the Results section, the globally unique identifier (GUID), currently in the form of a Life Science Identifier (LSID), should be listed under the new species name, for example: Anochetus boltoni Fisher sp. nov. urn:lsid:zoobank.org:act:B6C072CF-1CA6-40C7-8396-534E91EF7FBBAnother LSID for the manuscript itself should also appear within the Nomenclature statement. You will need to contact Zoobank (zoobank.org/About) to obtain a GUID (LSID). You should receive one LSID for your manuscript and a separate, unique LSID for the new species. B.
Please also insert the following text into the Methods section, in a sub-section to be called ""Nomenclatural Acts"": The electronic edition of this article conforms to the requirements of the amended International Code of Zoological Nomenclature, and hence the new names contained herein are available under that Code from the electronic edition of this article. This published work and the nomenclatural acts it contains have been registered in ZooBank, the online registration system for the ICZN. The ZooBank LSIDs (Life Science Identifiers) can be resolved and the associated information viewed through any standard web browser by appending the LSID to the prefix ""http://zoobank.org/"". The LSID for this publication is: urn:lsid:zoobank.org:pub: XXXXXXX. The electronic edition of this work was published in a journal with an ISSN, and has been archived and is available from the following digital repositories: PubMed Central, LOCKSS [author to insert any additional repositories]. All PLOS ONE articles are deposited in PubMed Central and LOCKSS. If your institute, or those of your co-authors, has its own repository, we recommend that you also deposit the published online article there and include the name in your article.Following a recent ruling by the International Commission on Zoological Nomenclature, electronic journals are now a valid format for publication of new zoological taxa. In order to ensure the valid publication of your new species, please be sure to include the updated version of Nomenclatural Acts (above). A complete explanation of our guidelines for publishing new species can be found on our website: http://www.plosone.org/static/guidelines#zoological. 4. Thank you for stating the following financial disclosure: "Michael Balke acknowledges support from the Deutsche Forschungsgemeinschaft (BA2152/27-1), project number: 496550039. In the initial phases of this project, Adrián Villastrigo was funded by the Alexander von Humboldt Foundation through a Humboldt Research Fellowship and by the Carl Friedrich von Siemens Foundation at SNSB-ZSM, and this stipend is greatly acknowledged here. Michael Balke acknowledges support from the EU SYNTHESYS program, projects FR-TAF 6972 and GB-TAF-6776, which supported this research during visits to Natural History Museum in London and Muséum national d’Histoire naturelle in Paris in 2017 to study historical type material." Please state what role the funders took in the study.  If the funders had no role, please state: ""The funders had no role in study design, data collection and analysis, decision to publish, or preparation of the manuscript."" If this statement is not correct you must amend it as needed. Please include this amended Role of Funder statement in your cover letter; we will change the online submission form on your behalf. 5. Thank you for stating the following in the Acknowledgments Section of your manuscript: "This work was made possible by a grant from the Alexander von Humboldt foundation under the Research Group Linkage Program (Evolution of the high Andean insect fauna project) which enabled the Colombian and German teams to closely work together and develop new partnerships on both sides. We are grateful for the generous support from the SNSB-Innovative scheme, funded by the Bayerisches Staatsministerium für Wissenschaft und Kunst (Project: “Geographische Isolation, Endemismus und Artbildungsprozesse bei Insekten in der hochmontanen Páramo Kolumbiens (und darüber hinaus)”)." We note that you have provided funding information that is not currently declared in your Funding Statement. However, funding information should not appear in the Acknowledgments section or other areas of your manuscript. We will only publish funding information present in the Funding Statement section of the online submission form. Please remove any funding-related text from the manuscript and let us know how you would like to update your Funding Statement. Currently, your Funding Statement reads as follows: "Michael Balke acknowledges support from the Deutsche Forschungsgemeinschaft (BA2152/27-1), project number: 496550039. In the initial phases of this project, Adrián Villastrigo was funded by the Alexander von Humboldt Foundation through a Humboldt Research Fellowship and by the Carl Friedrich von Siemens Foundation at SNSB-ZSM, and this stipend is greatly acknowledged here. Michael Balke acknowledges support from the EU SYNTHESYS program, projects FR-TAF 6972 and GB-TAF-6776, which supported this research during visits to Natural History Museum in London and Muséum national d’Histoire naturelle in Paris in 2017 to study historical type material." Please include your amended statements within your cover letter; we will change the online submission form on your behalf. 6. Your ethics statement should only appear in the Methods section of your manuscript. If your ethics statement is written in any section besides the Methods, please move it to the Methods section and delete it from any other section. Please ensure that your ethics statement is included in your manuscript, as the ethics statement entered into the online submission form will not be published alongside your manuscript. 7. We note that you have referenced (Balke et al., unpublished)which has currently not yet been accepted for publication. Please remove this from your References and amend this to state in the body of your manuscript: (ie “Balke et al. [Unpublished]”) as detailed online in our guide for authorshttp://journals.plos.org/plosone/s/submission-guidelines#loc-reference-style

Reviewers' comments:

Reviewer's Responses to Questions

**Comments to the Author**

1. Is the manuscript technically sound, and do the data support the conclusions?

Reviewer #1: Yes

Reviewer #2: Yes

2. Has the statistical analysis been performed appropriately and rigorously? 

Reviewer #1: Yes

Reviewer #2: Yes

3. Have the authors made all data underlying the findings in their manuscript fully available?

Reviewer #1: Yes

Reviewer #2: Yes

4. Is the manuscript presented in an intelligible fashion and written in standard English?

Reviewer #1: Yes

Reviewer #2: Yes

5. Review Comments to the Author

Reviewer #1: Review | Manuscript: PONE-D-24-13211

SUMMARY AND OVERALL IMPRESSION

The current study presents a novel integrative approach to resolve the taxonomy and population genetic structure using the genus Liodessus Guignot 1939 as a model system. Further, this work explores unique ecosystems, namely Páramo habitats that are good models for investigating population genetic structure due to intermittent geographical isolations and connectivity in the past. Therefore, the work will significantly contribute to advancing the field and will be of interest to the broad community.

GENERAL COMMENTS

Overall statistics: In the methods section, it will be helpful from the readers' point of view to describe statistics relevant to analyses performed. Similarly, please describe in detail the statistical results in the results section. It will be useful for replication and repetition of the study.

Population genetic structure: This work includes 147 specimens of L. quimbaya and 59 specimens of L. quillacinga to explore population genetic structure. Which test/tests were used to check data adequacy prior to analysis?

Correspondence analysis: It will be interesting to investigate the patterns of distribution based on wing morphology using the relevant high dimensional data reduction method, namely correspondence analysis. It’s a simple analysis based on frequency data to study species distribution based on vestigial and normal wings. Authors already have this data using a huge number of specimens that are reflected in the material examined section under taxonomy.

Trait mapping: As mentioned in the introduction section it will be fascinating to explore the relation between elevation gradient and wing size of Liodessus specimens. Have the authors observed any patterns of reduction in wing size based on elevation gradient? This trait can be further mapped on the phylogeny using Phytools in R.

Have authors also found a mixture of normal-winged and vestigial-winged specimens in the same population?

SPECIFIC COMMENTS

The specific comments are marked in the pdf file. Therefore, those are not listed here to avoid redundancy.

DECISION

Major revision

Reviewer #2: Congratulations on a well written and interesting piece of work, providing deeper taxonomic and biogeographic knowledge on predaceous diving beetles of the northern Andes as study system. Overall subspecies appear to be useful in this taxonomic group, however it seems in some cases they may not be that easy to detect based on morphology, which may pose difficulties for the average entomologist or biomonitor of ecosystems. This may be an intrinsic problem of the group, if some subspecies are in essence cryptic. I recommend in further publications, authors may provide recommendations to biomonitoring professionals and aquatic biologists interested in conservation on how to cope with difficult or cryptic subspecies complexes, either through user-friendly molecular protocols or lower morphological resolution alternatives.

6. PLOS authors have the option to publish the peer review history of their article (what does this mean?). If published, this will include your full peer review and any attached files.

Reviewer #1: **Yes: **Sayali D. Sheth

Reviewer #2: **Yes: **Atilano Contreras-Ramos

---

## [Author Response · Author response to Decision Letter 0]

1 Jul 2024

Dear Editor,

We are happy to see the positive feedback for our manuscript “Recent range expansion and lineage idiosyncratic population structure of Liodessus diving beetles in the high Andes (Coleoptera: Dytiscidae, Bidessini)”. We have addressed most comments, although some of them are further discussed in this letter. Regarding your recommendation of including some species delimitation methods, we have tested ASAP and PTP, which highlighted some unexpected results in the case of Liodessus quillacinga which are not consistent with the other evidence we used. For that reason, we opted to include that information as part of the supplementary material, and added comments in the results and discussion.

Regarding other comments made in your email, please let us clarify some of them:

4. Thank you for stating the following financial disclosure…

REPLY >>> The contributions made by Michael Balke were not only funding acquisition, but active work on it (as can be seen as been the first author). We participated in all steps, from field work, to specimen curation, experimental design, data analyses and discussion and manuscript writing. Do you require any specific statement within the financial disclosure?

5. Thank you for stating the following in the Acknowledgments Section of your manuscript…

REPLY >>> Thank you for letting us know about our mistake. Indeed, the support of some institutions was not clearly stated in the financial disclosure and was in the acknowledgements. A complete financial disclosure should be used as follows:

“Michael Balke acknowledges support from the Deutsche Forschungsgemeinschaft (BA2152/27-1), project number: 496550039. In the initial phases of this project, Adrián Villastrigo was funded by the Alexander von Humboldt Foundation through a Humboldt Research Fellowship and by the Carl Friedrich von Siemens Foundation at SNSB-ZSM, and this stipend is greatly acknowledged here.

Michael Balke acknowledges support from the EU SYNTHESYS program, projects FR-TAF 6972 and GB-TAF-6776, which supported this research during visits to Natural History Museum in London and Muséum national d’Histoire naturelle in Paris in 2017 to study historical type material.

This project was also generously supported by a grant from the Alexander von Humboldt Foundation under the Research Group Linkage Program (Evolution of the high Andean insect fauna project) which enabled the Colombian and German teams to closely work together and develop new partnerships on both sides. We are also grateful for the generous support from the SNSB-Innovative scheme, funded by the Bayerisches Staatsministerium für Wissenschaft und Kunst (Project: “Geographische Isolation, Endemismus und Artbildungsprozesse bei Insekten in der hochmontanen Páramo Kolumbiens (und darüber hinaus)”).”

Reviewer #1 >>> Review | Manuscript: PONE-D-24-13211

SUMMARY AND OVERALL IMPRESSION

The current study presents a novel integrative approach to resolve the taxonomy and population genetic structure using the genus Liodessus Guignot 1939 as a model system. Further, this work explores unique ecosystems, namely Páramo habitats that are good models for investigating population genetic structure due to intermittent geographical isolations and connectivity in the past. Therefore, the work will significantly contribute to advancing the field and will be of interest to the broad community.

REPLY >>> Thank you very much for your positive opinion on our research. It is indeed a fantastic study system, and we are confident it will continue to yield great insights into the macro- and microevolutionary factors influencing speciation.

GENERAL COMMENTS

Reviewer >>> Overall statistics: In the methods section, it will be helpful from the readers' point of view to describe statistics relevant to analyses performed. Similarly, please describe in detail the statistical results in the results section. It will be useful for replication and repetition of the study.

REPLY >>> As you mentioned, we recognize the needs to provide comprehensive methodological details to facilitate replication. We have described all software used and the parameters required for replications (e.g., specific alignment settings, tree-building software with algorithm, haplotype phasing, Fst calculation, …). We believe the information provided in the manuscript is sufficient for replication. The only “missing information” pertains to the visualization of the Fst values using Python 3 and the Seaborn library, which is just a visualization method rather than a proper methodology.

Reviewer >>> Population genetic structure: This work includes 147 specimens of L. quimbaya and 59 specimens of L. quillacinga to explore population genetic structure. Which test/tests were used to check data adequacy prior to analysis?

REPLY >>> As stated in the methodology (lines 145 to 163), we followed a standard pipeline to analyze molecular sequences. Specifically, for the population genetic structure analyses, we used two approaches: visual exploration of the population structure by haplotype networks and the use of the fixation index (Fst), a common measure of population differentiation. Prior to running these methods, we visually inspected the chromatograms (now mentioned in lines 137-138) and performed another visual inspection of the alignments to detect potential amplification or sequencing errors (e.g., internal stop codons).

Reviewer >>> Correspondence analysis: It will be interesting to investigate the patterns of distribution based on wing morphology using the relevant high dimensional data reduction method, namely correspondence analysis. It’s a simple analysis based on frequency data to study species distribution based on vestigial and normal wings. Authors already have this data using a huge number of specimens that are reflected in the material examined section under taxonomy.

Trait mapping: As mentioned in the introduction section it will be fascinating to explore the relation between elevation gradient and wing size of Liodessus specimens. Have the authors observed any patterns of reduction in wing size based on elevation gradient? This trait can be further mapped on the phylogeny using Phytools in R.

Have authors also found a mixture of normal-winged and vestigial-winged specimens in the same population?

REPLY >>> Thank you for your insight on a potential new analysis regarding wing morphology and adaptation to altitude, which we also are very excited about. Vestigial wings are usually found in high altitudes, as they are a well-known adaptation to cope with extreme conditions such as strong winds. Most specimens in Páramo or Puna ecosystems have vestigial wings. To analyze wing morphology in detail, it would be necessary to incorporate multiple species from various altitudes, not just those inhabiting high altitudes. Unfortunately, we cannot include such an extensive analysis in this manuscript. However, we are working on a broader study with comprehensive sampling and a calibrated phylogenomic analysis of most known Liodessus species to answer exactly these questions you asked about speciation and adaptability to high elevations in Liodessus diving beetles.

Reviewer >>> Line 31: Please change the makred key words. Because these words already appear in the title.

REPLY >>> Thank you for the suggestions. We have modified the key words to avoid repetitions with the title.

Reviewer >>> Line 89: Please enlist the species and subspecies considered for the current work.

REPLY >>> Thank you for your suggestion. That information is included in the revised checklist of the paper

Updated checklist of Liodessus from Colombia and Ecuador

(* taxa treated herein)

Liodessus bogotensis bogotensis Guignot, 1953 (Colombia)

Liodessus b. almorzadero Balke et al., 2023 (Colombia)

Liodessus b. chingaza Balke et al., 2023 (Colombia)

Liodessus b. lacuniviridis Balke et al., 2020 (Colombia)

Liodessus b. matarredonda Balke et al., 2023 (Colombia)

Liodessus b. sumapaz Balke et al., 2023 (Colombia)

Liodessus picinus Balke et al., 2021 (Colombia)

Liodessus quillacinga quillacinga Megna, Hendrich & Balke, 2019 (Colombia) *

Liodessus quillacinga cochaensis Megna, Hendrich & Balke, 2019 (Colombia) *

Liodessus quillacinga cumbalis Megna, Hendrich & Balke, 2019 (Colombia) *

Liodessus quillacinga ecuadoriensis ssp. nov. (Ecuador) *

Liodessus quimbaya quimbaya Megna, Hendrich & Balke, 2019 (Colombia) *

Liodessus quimbaya azufralis Megna, Hendrich & Balke, 2019 stat. nov. (Colombia, Ecuador) *

Liodessus quimbaya paletara ssp. nov. (Colombia) *

Liodessus riveti Peschet, 1923 (Ecuador) *

Liodessus santarosita Balke et al., 2023 (Colombia)

Reviewer >>> Line 124 and 135: Please elaborate on statistics used.

REPLY >>> We have made minor modifications to this section to clarify the methods used. Geneious incorporates a tool to generate neighbor-joining trees using various genetic distance models. More detailed information is provided in the following section (lines 133 to 143).

Reviewer >>> Line 149: How the groups of populations were made?

REPLY >>> By “population”, we refer to the general definition of individuals of the same species coexisting in the same location and time. As shown in the Fst figures, we maintained the same locality codes as in the manuscript to facilitate comparisons among populations and not only among subspecies, affectively assessing population genetic structure. 

Reviewer >>> Line 160: Please mention/describe statistical results. Also Line 397.

REPLY >>> We agree that mentioning the main statistical results is essential in any scientific manuscript. For that reason, we had included a section on the molecular evidence, where we describe the main results of our analyses.

Reviewer >>> Table 1: Are these two populations of the same subspecies? Or subspecies quimbaya is repeated mistakenly?

REPLY >>> Thank you for pointing out this issue. It was indeed a repeated word that should not be there. We have corrected the manuscript accordingly.

Reviewer >>> Line 453: Has phylogeography of the genus being worked out? Are there any evidences based on the molecular clock about the origin of the genus in such a dynamic ecosystems? Please discuss.

REPLY >>> We are currently working on a comprehensive study of Liodessus, including most of the described species and some new independent genetic lineages, using whole-genome sequencing and calibrations. However, we are still in the dataset-building phase, and such a work will take some time before it is ready for review. In the meantime, we have been addressing species inconsistencies across Liodessus to use that information as a baseline for a comprehensive analysis.

Reviewer >>> Line 467-469: Please mention a reference for this sentence or simply remove. Based on the data presented, the current study does not conduct diversity analyes.

REPLY >>> The sentence was a continuation of the previous one, referring to the pattern found by Calpe-Anaguano et al. in their study of bird species in the Páramo. We have explicitly incorporated a reference in that sentence to avoid any misunderstanding.

Reviewer >>> Reviewer #2: Congratulations on a well written and interesting piece of work, providing deeper taxonomic and biogeographic knowledge on predaceous diving beetles of the northern Andes as study system. Overall subspecies appear to be useful in this taxonomic group, however it seems in some cases they may not be that easy to detect based on morphology, which may pose difficulties for the average entomologist or biomonitor of ecosystems. This may be an intrinsic problem of the group, if some subspecies are in essence cryptic. I recommend in further publications, authors may provide recommendations to biomonitoring professionals and aquatic biologists interested in conservation on how to cope with difficult or cryptic subspecies complexes, either through user-friendly molecular protocols or lower morphological resolution alternatives.

REPLY >>> Thank you very much for your kind comments. Indeed, morphological convergence is likely an issue in the Liodessus species inhabiting high altitudes in the Andes. To fully understanding how these beetles diversified in such extreme habitats and assess their conservation status, molecular evidence is required. We are aware of the difficulties of dealing with (almost) cryptic species, and we will take your recommendations on biomonitoring and conservation into account for future studies.

Reviewer >>> Line 59-60:

REPLY >>> We did not specify a country because Páramo and Puna are the names of high-altitude plateaus in the Andes, with Páramo most frequently used in northern Andes and Puna in Central and Southern Andes.

Reviewer >>> Line 262: is there any additional distinguishing feature from geographic locality?

REPLY >>> No. We intend to highlight that the subspecies is characterized by its geographic location.

---

## [Decision Letter · Decision Letter 1]

30 Jul 2024

Recent range expansion and lineage idiosyncratic population structure of Liodessus diving beetles in the high Andes (Coleoptera: Dytiscidae, Bidessini)

PONE-D-24-13211R1

Dear Dr. Villastrigo,

We’re pleased to inform you that your manuscript has been judged scientifically suitable for publication and will be formally accepted for publication once it meets all outstanding technical requirements.

Kind regards,

Łukasz Kajtoch, Ph.D.

Academic Editor

PLOS ONE

Additional Editor Comments (optional):

Reviewers' comments:

Reviewer's Responses to Questions

**Comments to the Author**

1. If the authors have adequately addressed your comments raised in a previous round of review and you feel that this manuscript is now acceptable for publication, you may indicate that here to bypass the “Comments to the Author” section, enter your conflict of interest statement in the “Confidential to Editor” section, and submit your "Accept" recommendation.

Reviewer #1: All comments have been addressed

Reviewer #2: All comments have been addressed

2. Is the manuscript technically sound, and do the data support the conclusions?

Reviewer #1: Yes

Reviewer #2: Yes

3. Has the statistical analysis been performed appropriately and rigorously? 

Reviewer #1: Yes

Reviewer #2: Yes

4. Have the authors made all data underlying the findings in their manuscript fully available?

Reviewer #1: Yes

Reviewer #2: Yes

5. Is the manuscript presented in an intelligible fashion and written in standard English?

Reviewer #1: Yes

Reviewer #2: Yes

6. Review Comments to the Author

Reviewer #1: (No Response)

Reviewer #2: Thanks for taking care of my comments, indeed, my concern is more on future grounds regarding work in the study area and conservation efforts for these interesting ecosystems and their biota, so that identification of the beetle subspecies may be accessible to the biomonitoring entomologist or conservation biologist, given the cryptic nature of some of the subspecies. Yet, justification on molecular grounds is valid and well sustained. Again, congratulations on a nice piece of work on this complex taxonomic group of aquatic beetles.

7. PLOS authors have the option to publish the peer review history of their article (what does this mean?). If published, this will include your full peer review and any attached files.

Reviewer #1: **Yes: **Sayali D. Sheth

Reviewer #2: No
